# Bystander hyperactivation of preimmune CD8+ T cells in chronic HCV patients

Cécile Alanio[1,2,3], Francesco Nicoli[4,16], Philippe Sultanik[1,3], Tobias Flecken[5], Brieuc Perot[1,3], Darragh Duffy[1,2,3], Elisabetta Bianchi[6], Annick Lim[7], Emmanuel Clave[8], Marit M van Buuren[9], Aurélie Schnuriger[10], Kerstin Johnsson[11], Jeremy Boussier[1,2,3], Antoine Garbarg-Chenon[10], Laurence Bousquet[12], Estelle Mottez[2], Ton N Schumacher[9], Antoine Toubert[8], Victor Appay[4,16], Farhad Heshmati[13], Robert Thimme[5], Stanislas Pol[12], Vincent Mallet[12], Matthew L Albert[1,2,3]*

[1]Unités de Recherche Internationales Mixtes Pasteur, Institut Pasteur, Paris, France; [2]Centre d'Immunologie Humaine, Institut Pasteur, Paris, France; [3]Immunobiology of Dendritic Cells, Institut Pasteur, Paris, France; [4]Sorbonne Universités, UPMC Univ Paris 06, DNU FAST, CR7, Centre d'Immunologie et des Maladies Infectieuses (CIMI-Paris), Paris, France; [5]The University Medical Center Freiburg, Department of Internal Medicine II, Albert-Ludwigs-Universität, Freiberg, Germany; [6]Immunoregulation Unit, Institut Pasteur, Paris, France; [7]Plateforme d'Immunoscope, Institut Pasteur, Paris, France; [8]Hôpital Saint-Louis, Assistance publique - hôpitaux de Paris, Paris, France; [9]Department of Immunology, The Netherlands Cancer Institute, Amsterdam, The Netherlands; [10]Laboratoire de virologie, Hôpital Armand-Trousseau, Assistance publique - hôpitaux de Paris, Paris, France; [11]Mathematics, Faculty of Engineering, Lunds University, Lund, Sweden; [12]APHP, Université Paris Descartes, Paris, France; [13]EFS, Hôpital Cochin, Paris, France; [14]Centre d'Immunologie et des Maladies Infectieuses, University Pierre et Marie Curie, Paris, France; [16]U1135, INSERM, CIMI-Paris, Paris, France

*For correspondence: albertm@pasteur.fr

**Abstract** Chronic infection perturbs immune homeostasis. While prior studies have reported dysregulation of effector and memory cells, little is known about the effects on naïve T cell populations. We performed a cross-sectional study of chronic hepatitis C (cHCV) patients using tetramer-associated magnetic enrichment to study antigen-specific inexperienced CD8+ T cells (i.e., tumor or unrelated virus-specific populations in tumor-free and sero-negative individuals). cHCV showed normal precursor frequencies, but increased proportions of memory-phenotype inexperienced cells, as compared to healthy donors or cured HCV patients. These observations could be explained by low surface expression of CD5, a negative regulator of TCR signaling. Accordingly, we demonstrated TCR hyperactivation and generation of potent CD8+ T cell responses from the altered T cell repertoire of cHCV patients. In sum, we provide the first evidence that naïve CD8+ T cells are dysregulated during cHCV infection, and establish a new mechanism of immune perturbation secondary to chronic infection.

## Introduction

Functional impairments of CD8+ T cells have been characterized in several persistent viral infections, including human immunodeficiency virus (HIV) and hepatitis C virus (HCV) infection in humans, simian immunodeficiency virus (SIV) infection in macaques, and lymphocytic choriomeningitis virus (LCMV)

**eLife digest** Long-lasting or "chronic" infections massively perturb the immune system as a way to favor their own growth. In particular, they can stop T cells – a subtype of immune cells that help to destroy viruses – from working well. For example, HIV and hepatitis C viruses can overwork T cells and cause them to die. This can make individuals vulnerable to other infections.

In healthy people, T cells that have participated in the fight against particular infections continue to live to provide a memory of those past infections. Groups of "naïve" T cells that have not yet encountered an infected cell also patrol the body, ready to respond to infections by a new virus. There are relatively few virus-specific naïve T cells in the body, so until recently it has been hard to study them. As a result, researchers know little about how these cells are affected by long-lasting infections, and whether chronic infection affects our capacity to fight unrelated infections.

Alanio et al. have now used a highly sensitive technique to compare naïve T cells found in the blood of three groups of people: those with chronic hepatitis C infections, those who have been cured of a chronic hepatitis C infection, and healthy people. This revealed that the naïve T cells are negatively affected by chronic hepatitis C infections, and become hypersensitive: they get easily overexcited, which can lead to their death. This compromises the immune defenses at the moment they are most needed.

Closer inspection showed that the naïve T cells of patients with hepatitis C are hypersensitive because they have less of a protein called CD5 on their surface. This protein acts as a natural brake for the T cells, and thus having less results in the T cells mounting stronger immune responses. Although this might be beneficial when fighting certain infections, this may also account for conditions where T cells attack healthy tissues.

Finally, Alanio et al. found evidence that people who have been cured of a chronic hepatitis C infection recover a healthy set of naïve T cells within two years. Treating patients as soon as an infection is diagnosed therefore has several benefits: as well as clearing the virus, this will reset the immune system balance and reduce the damage that hyperactive immune cells cause to the body.

The results also have implications for vaccinations, which work by pushing naïve T cells to arm themselves against a particular virus. The discovery that naïve T cells are hypersensitive in patients with hepatitis C suggests that we may need a distinct strategy for efficiently vaccinating these patients. It is indeed possible that standard vaccines – tested in groups of healthy people – may result in unexpected and unwanted immune responses in individuals with hepatitis C.

These open questions will be addressed in further studies. It will also be of interest to know if other chronic viruses have the same ability to alter the activity of naïve T cells.

infection in mice (*Ahmed and Rouse, 2006*). In particular, it has been shown that chronic infection skews memory/effector CD8[+] T cell differentiation (*Stelekati and Wherry, 2012*), and drives virus-specific CD8[+] T cells towards an « exhausted » phenotypic state, as marked by high expression of the programmed cell death-1 (PD-1) molecule (*Kim and Ahmed, 2010*). Chronic infections have also been reported to impair immune responses to unrelated infectious microbes in mouse models (*Stelekati and Wherry, 2012*; *Richer et al., 2013*), as well as in humans infected with HCV (*Park and Rehermann, 2014*). This phenomenon correlates with a interferon (IFN) stimulated gene (ISG) transcriptional signature, suggesting an indirect effect of systemic type I IFN secondary to innate immune activation (*Stelekati et al., 2014*). Following from these observations, we hypothesized that chronic infection may alter the T cell preimmune repertoire, which plays an important role in shaping the adaptive immune responses (*Jenkins and Moon, 2012*). Employing a newly validated approach for the study of low-frequency ($< 10^{-5}$) antigen-specific T cells (*Alanio et al., 2010*), we evaluated this prediction in patients with chronic viral infection of the liver.

The α/β T cell preimmune repertoire is defined as the set of mature but antigen inexperienced lymphocytes that circulate in blood and secondary lymphoid organs, ready to be activated by cognate high-affinity peptide-class I MHC (pMHC) complexes (*Jenkins et al., 2010*). They are maintained in the periphery by survival factors such as IL-7, as well as transient contacts with low affinity non-cognate pMHC complexes (*Sprent and Surh, 2011*). Over the last decade, studies using newly-developed tetramer-enrichment assays - sensitive enough to detect and track antigen-specific

populations prior to immunization - have provided new insights into the impact of preimmune repertoire heterogeneity (*Jenkins et al., 2010*). First, the number of antigen-specific T cells (i.e. precursor frequency) is not equivalent across inexperienced populations, with the absolute number positively correlating with the magnitude of responses that are induced upon priming (*Obar et al., 2008*; *Moon et al., 2007*; *Kwok et al., 2012*; *Schmidt et al., 2011*; *Kotturi et al., 2008*; *Tan et al., 2011*). Second, antigen-inexperienced CD4[+] and CD8[+] T cell populations contain variable proportions of memory-phenotype (MP) cells (*Legoux et al., 2010*; *Su et al., 2013*). These cells have been explained in the literature as a result of cross-reactivity or homeostatic proliferation (*Sprent and Surh, 2011*). Cross-reactivity is now recognized as an essential feature of the T-cell receptor (TCR) / MHC interaction (Mason et al., 1998), and a major determinant of virus-specific MP cells in the CD4[+] T cell repertoire of unexposed healthy donors (*Su et al., 2013*). Alternatively, homeostatic proliferation may occur in settings of lymphopenia (*Jones et al., 2013*). Finally, differential CD5 expression by antigen-specific T cell populations has been shown to dictate clonal recruitment and expansion (*Fulton et al., 2015*; *Tabbekh et al., 2013*). To date, the impact of non-heritable influences such as human chronic viral infection on the quantitative and qualitative aspects of the preimmune repertoire remains unknown.

In our study, we focused on patients with chronic hepatitis C virus infection (cHCV), which show CD8[+] T cell dysfunction (*Park and Rehermann, 2014*; *Rehermann and Nascimbeni, 2005*). In particular, HCV-specific responses are typically (i) weak – both in term of numbers and function, (ii) of low avidity, and (iii) blocked in their differentiation into central memory cells, despite the availability of cognate pMHC complexes (*Park and Rehermann, 2014*). cHCV is to date the only chronic viral infection that can be cured, offering the unique possibility to interrogate the reversibility of immune perturbations post-viral clearance (*Pol et al., 2013*). Herein, we applied the highly sensitive tetramer-associated magnetic enrichment (TAME) technique for investigating at the antigen-specific level the impact of chronic viral hepatitis infection on the CD8 T cell preimmune repertoire (*Alanio et al., 2010*). Although precursor frequencies were similar to healthy controls, we observed significant impairments of the preimmune repertoire in cHCV patients. Inexperienced T cell populations contained increased proportions of MP cells. This correlated with naïve-phenotype CD8[+] T cells having lower surface expression of CD5, which accounted for a lower threshold for TCR signaling and the generation of potent immune responses from cHCV patients. Importantly, the positive effect of chronic infection on naïve T cell recruitment into immune responses is transient, as cHCV patients who clear their virus following successful therapy (referred to as Sustained Virologic Responders or SVR) can experience a reversion towards a healthy naïve T cell repertoire within 2 years. These data provide the first evidence for chronic infection resulting in the bystander dysregulation of the antigen-specific preimmune repertoire in humans, and highlight the added benefit of early viral clearance in patients with chronic HCV infection.

## Results

### Perturbed naïve CD8[+] T cell repertoire during chronic infection

To test the hypothesis that chronic viral infection perturbs preimmune repertoire homeostasis, we evaluated the influence of cHCV infection on the phenotype of circulating CD8[+] T cells. 29 cHCV and 37 Sustained Virologic Responders (SVR, *i.e.* patients achieving clearance of the virus after therapy) patients were included in the study (*Table 1*). 62% of the chronic and 100% of the SVR patients received at least one anti-HCV treatment (of those treated, 69% received conventional IFN-ribavirin bitherapy, 31% IFN + direct antiviral agent (DAA), and IFN-free DAA combination therapy alone in the case of a single SVR patient). Healthy donors from the blood bank were included as controls. Total lymphocyte numbers were within the normal range for all tested patients (median 2.2 +/- 0.6 G/l). Within the CD3[+] lymphocyte population, we observed similar percentages of circulating CD8[+] T cells (*Figure 1—figure supplement 1*). However, absolute numbers of CD3[+] were significantly increased in our cohort of cHCV (KW p<0.0001), translating into increased absolute numbers of CD8[+] T cells in cHCV patients (KW p=0.0002) (*Figure 1—figure supplement 2*). We further subsetted the CD8[+] T cells according to their surface expression of CD45RA and CD27. Based on prior studies (*Alanio et al., 2010*; *De Rosa et al., 2001*) and our confirmatory experiments using 5 phenotypic markers for naïve or memory T cells, we determined that co-expression of high levels of

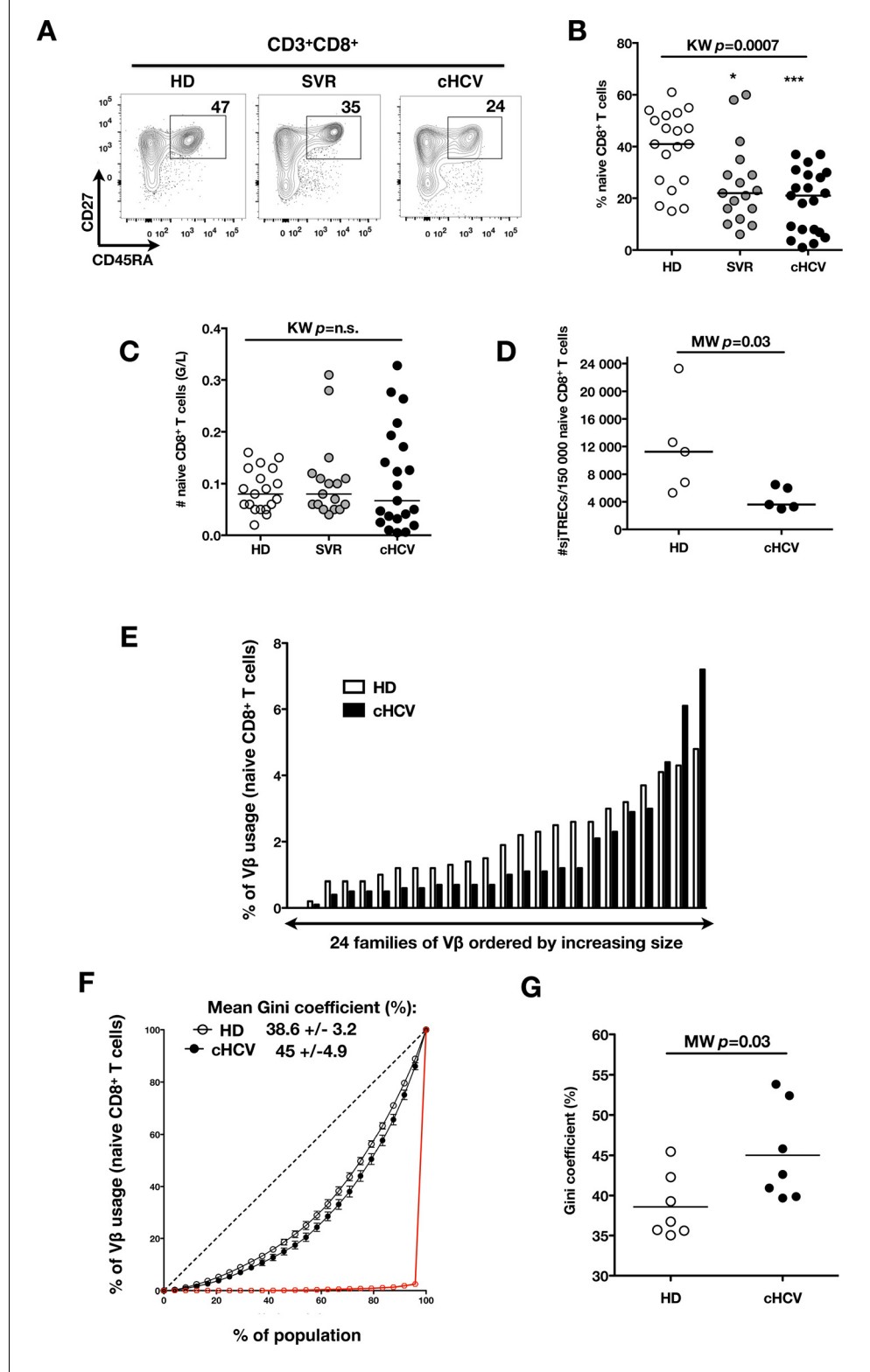

**Figure 1.** Perturbed naïve CD8+ T cell repertoire during chronic HCV infection. Percentages and absolute numbers of CD3+ and CD3+CD8+ cells in Healthy Donors (HD), Sustained Virologic Responder (SVR), and chronic HCV (cHCV) patients are provided in *Figure 1—figure supplement 1* and *2*. (A) Representative examples of CD45RA+CD27+ naïve CD8+ T cell compartment in the three donor subsets. FACS plots are gated on Live CD3+CD8+ cells. Validation of CD45RA/CD27 gating strategy for identifying naïve CD8+ T cells in cHCV patients is provided in *Figure 1—figure supplement 3*. (B) Percentages of naïve CD8+ T cells in the three donor subsets. (C) Absolute numbers (G/L) of naïve CD8+ T cells in HD, SVR, and cHCV patients. ns (not

*Figure 1 continued on next page*

*Figure 1 continued*

significant, p>0.05, *(p≤0.05), **(p≤0.01), and ***(p≤0.001) refer to Dunn's multiple comparison test of each subset toward HD. (**D**) Normalized numbers of sjTRECs per 150,000 naïve CD8[+] in HD and cHCV samples. Normalized numbers of sjTRECs per total CD8[+] T cells are provided in *Figure 1—figure supplement 4*. (**E**) Representative example of the distribution of 24 FACS-screened Vβ families in naïve CD8[+] T cells from one HD and one cHCV sample. Families are ordered by increasing size in both individuals. (**F**) Lorenz curves representing the cumulative distribution of % of usage of 24 FACS-screened Vβ families from 7 HD and 7 cHCV patients. Mean Gini coefficients and standard deviations are indicated. Red line indicates an extreme example of an unequal distribution, observed in the case of a T-cell lymphoma where >90% of the TCR repertoire is explained by one particular Vβ chain. (**G**) Individual Gini coefficients of all tested samples are represented for HD and cHCV subgroups.

The following figure supplements are available for figure 1:

**Figure supplement 1.** Comparable proportions of CD8[+] T cells circulate in cHCV patients and HD.

**Figure supplement 2.** Increased absolute numbers of CD3[+] and CD8[+] T cells in cHCV and SVR patients.

**Figure supplement 3.** Validation of CD45RA/CD27 gating strategy for identifying naïve CD8[+] T cells in cHCV patients.

**Figure supplement 4.** Decreased number of sjTRECs in total CD3[+]CD8[+] T cells from cHCV patients.

CD45RA and CD27 were sufficient to classify naïve T cells in both HD and cHCV patients (*Figure 1—figure supplement 3*). Decreased percentages of naïve CD8[+]T cells have previously been reported in cHCV (*Shen et al., 2010*). Here, we confirmed these findings in age- and CMV- matched chronically infected patients (KW p=0.0007, *Figure 1A,B*). Interestingly, we found that after correcting for the higher CD8[+] T cell numbers in cHCV patients, the absolute numbers of naïve CD8[+] T cells were within the normal range as determined by the study of healthy donors (*Figure 1C*). We therefore interpreted the lower proportion of naïve T cells to simply be a result of an expansion of the memory cell compartment.

To directly test this prediction, we isolated CD8[+] T cells and measured the frequency of signal joint TCR excision circles (sjTREC), by-products of TCR rearrangement, and previously validated as a measure of thymic production (*Rehermann and Nascimbeni, 2005*; *Clave et al., 2009*). Confirming previous studies, we found a significant decrease in sjTREC content of CD8[+] T cells (MW p=0.01, *Figure 1—figure supplement 4*). To address the bias due to differential naïve T cell number, we isolated CD45RA[+]/CD27[+] naïve CD8[+] T cells and assessed sjTREC frequencies. Surprisingly, we also observed within the naïve compartment a significantly lower sjTREC content in cHCV patients as compared to HD (MW p=0.03, *Figure 1D*). To further characterize this phenotype, we assessed the Vβ distribution within the naïve repertoire of cHCV patients. cHCV patients showed a biased repertoire with increased representation of selected Vβ families. A representative example of Vβ usage plotted as percentage across the 24 tested families, and ordered by increasing size from one cHCV patient and one HD is shown (*Figure 1E*). To compare distributions, Lorenz curves were constructed as a graphical representation of the diversity of the repertoire (*Figure 1F*). Inequality measurements in the Vβ distribution, comparing cHCV patients to HD, indicated proportions of naïve T cells being

**Table 1.** Donors included in the study.

| All donors | cHCV n=29 | SVR n=37 | HD n=25 |
|---|---|---|---|
| Male, n (%) | 16 (55) | 21 (57) | 12 (48) |
| Age, years, median (IQR1-3) | 48 (42-55) | 48 (44-58) | 38 (31-46) |
| IgG anti-CMV positive, n (%) | 14 (48) | 24 (51) | 11 (44) |
| Cirrhosis, n (%) | 5 (17) | 8 (22) | na |
| Treatment experienced, n (%) | 18 (62) | 37 (100) | na |
| Treatment (n per type: 0/1/2/3) | 11/12/6/0 | 0/26/10/1 | na |
| Delay post-treatment, years, median (IQR1-3) | 3.8 (3.4-4.2) | 1.7 (0.9-3.2) | na |

altered in their Vβ usage. In brief, for a given percentage (x) of the 24 Vβ chains, Lorenz curves indicate the proportion of the T cell population that have Vβ chains among the 24 * x% least abundant ones. An equal distribution is represented as the dotted line. By contrast, an extreme, unequal distribution is shown in red, as in the case of a T-cell lymphoma where >90% of the TCR repertoire is explained by one particular Vβ chain (red line). We included Gini coefficient as a numeric measure of Lorenz curve's based observations. It corresponds to the ratio of the area between the line representing equal use of all Vβ chains (dotted line) and the observed Lorenz curve to the total area below the line representing equal use. The higher the coefficient, the more unequal is the distribution. In line with our observation, we found Gini coefficients increased in cHCV patients (M-W p=0.03, see Material and Methods for details of calculation) (*Figure 1G*). These data support an overall perturbed naïve CD8$^+$ T cell repertoire in cHCV patients, with increased peripheral expansion of selected populations.

## MP Mart1-specific CD8$^+$ T cells during chronic infection may be reversed by viral clearance

To evaluate more precisely the impact of these perturbations on antigen-specific populations, we applied recently developed strategies to detect, quantify and phenotype rare inexperienced antigen-specific CD8$^+$ T cells (*Klenerman and Thimme, 2012*; *Alanio et al., 2013*). Specifically, we utilized TAME to enumerate and subdivide Mart1-specific T cell populations. While similar absolute numbers of Mart1-specific CD8$^+$ T cells were observed in our respective study groups (*Figure 2A,B*), SVR and cHCV patients showed a more differentiated phenotype (*Figure 2C*), defined by fewer CD45RA$^+$/CD27$^+$ and increased proportions of memory-phenotype (MP) cells (KW *p*<0.0001, *Figure 2D*). Of note, these MP cells were mostly of central-memory (CD45RA-CD27+) phenotype (*Figure 2—figure supplement 1*). Also, when considering only naïve-phenotype Mart1-specific cells, precursor frequencies were still comparable across the different study groups (*Figure 2—figure supplement 2*). We were able to purify sufficient numbers of Mart1-specific naïve- and memory- phenotype CD8$^+$ T cells from one HCV patient to perform an immunoscope analysis on the Vβ chain usage (*Figure 2E*). In line with our data in bulk T cells populations (*Figure 1*), we observed a restricted repertoire of Mart1-specific naïve T cells, with evidence of an expanded Vβ clonotype in memory cells. These data argue in favor of MP cells being the progeny of a perturbed naïve T cell repertoire. Although they could be expanded in response to either specific or non-specific signals, we favor the latter hypothesis based on prior knowledge of Mart1 antigen pattern of expression (*Pittet et al., 1999*).

We next extended our observations to other antigen specificities by using four additional multimers (hTERT1$_{572-580}$, human CMV pp65$_{495-503}$, Ebola NP$_{202-210}$ (*Sundar et al., 2007*), HIV-1 Gag p17$_{77-85}$) that are expected to detect inexperienced self- and virus-specific CD8$^+$ T cell populations in tumor-free, CMV-, Ebola- and HIV- seronegative individuals. Here again, we found high proportions of T cells with a memory-phenotype in both self (Mart1- and hTERT- specific) and viral (CMV-, Ebola- and HIV- specific) antigen-inexperienced populations of cHCV patients as compared to healthy donors (representative plots are shown in *Figure 3A and B*; and combined results from 2–6 individuals per group in *Figure 3C*; self-specific: KW *p*<0.001; non-self-specific: KW *p*=0.009). When subsetted using CD45RA and CD27 phenotypic markers, the MP cells found in cHCV patients were preferentially of CD45RA$^-$CD27$^+$ central memory phenotype (*Figure 3—figure supplement 1*).

We next compared cHCV patients to those who achieved viral clearance. Sequential samples (available from five patients who achieved cure) suggested that immune restoration of the naïve compartment is possible (positive time dependency p-value *p*=0.03, *Figure 4A and B*). These patients were all treated by IFN-RBV biotherapy (*n* = 3), or triple therapy that included an NS3 inhibitor (*n* = 2, patients S2 and S12). Testing our observation in our cross-sectional cohort, we replicated our findings, showing a statistically significant recovery of naïve antigen-specific CD8$^+$ T cells as a function of time (MW *p*=0.04; *Figure 4C*). These results indicate that the differentiated cells within the perturbed repertoire of cHCV patients are a reflection of active HCV infection, and likely not a result of cross-reactivity or true memory T cell differentiation. Together the results in *Figures 1– 4* highlight an overall perturbation of the preimmune CD8$^+$ T cell compartment during active cHCV infection.

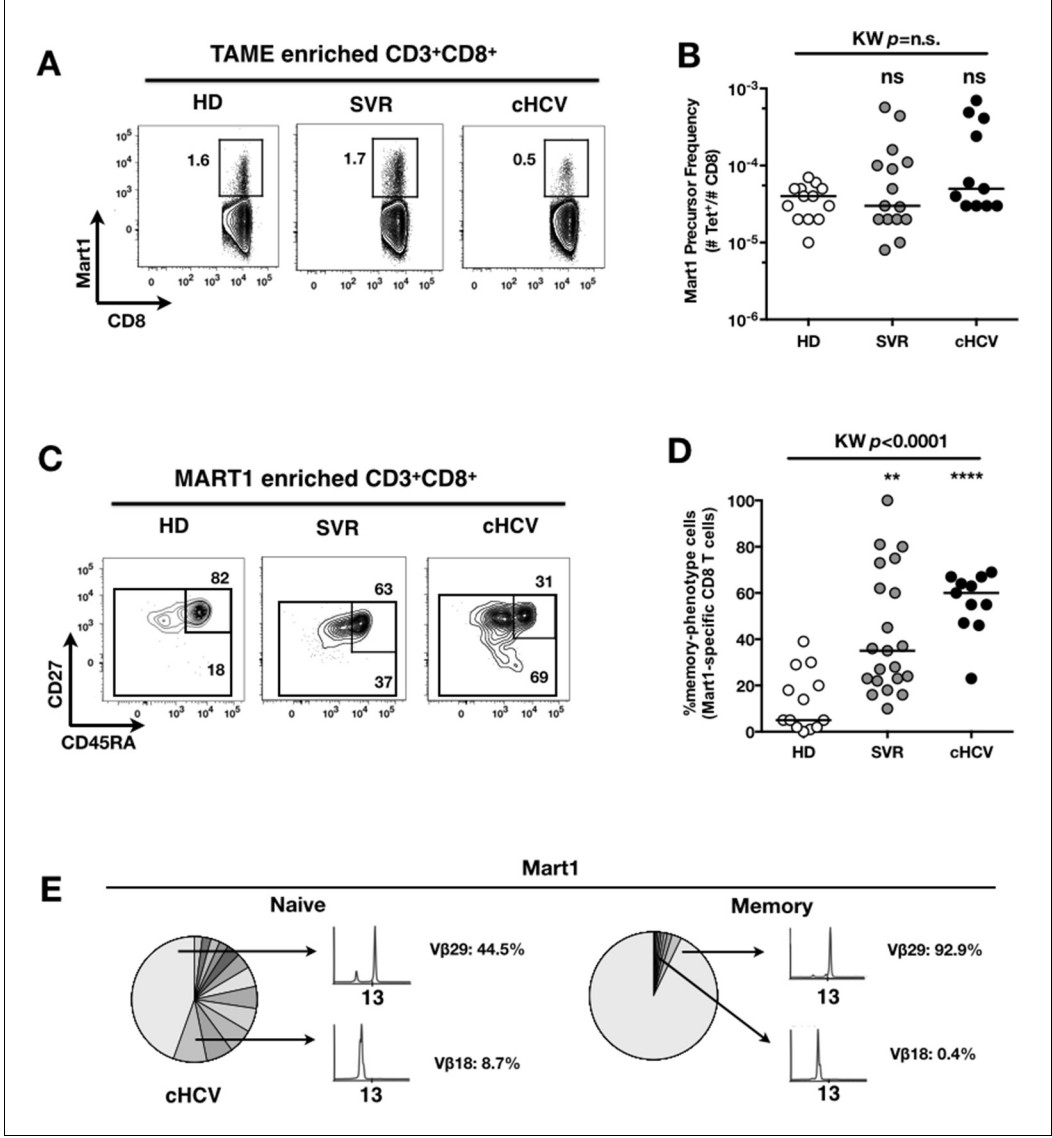

**Figure 2.** Peripheral differentiation of Mart1-specific CD8+ T cells during cHCV infection. (**A**) Representative examples of Mart1-specificCD8+ T cellpopulations in HD, SVR, cHCV patients. FACS plots are gated on TAME-enriched LiveFSC^lo^SSC^lo^CD3+CD8+ PBMCs. (**B**) Precursor frequency of Mart1-specific cellsin the three donor subsets. Precursor frequency of naïve-phenotype Mart1-specific cellsis provided in *Figure 2—figure supplement 1*. (**C**) Representative examples of the CD45RA/CD27 phenotype of TAME-enriched Mart1-specificpopulations in patients subsets as in A. **D**/ Percentages of memory-phenotype (MP) cells in Mart1-specific populations in the three donor subsets. Further subsetting of MP inexperienced T cells into CD45/CD27-based T cell differentiation phenotype is provided in *Figure 2—figure supplement 1*. E/ Immunoscope profile of naïve and memory Mart1-specific populations FACS-sorted from one cHCV patient.

The following figure supplements are available for figure 2:

**Figure supplement 1.** CD45RA/CD27-based subsetting of Mart1-specific T cells enriched from HD, SVR, cHCV.

**Figure supplement 2.** Precursor frequency of Mart1 naïve-phenotype cellsin the three donor subsets.

## Decreased expression of CD5 on naïve CD8+ T cells associates with TCR hypersensitivity in cHCV patients

To establish a mechanistic understanding of our findings, we considered the key homeostatic factors governing maintenance of the naïve CD8+ T cell compartment (*Jenkins et al., 2010*; *Ho and Hsue, 2009*; *Hazenberg et al., 2000*). We hypothesized that an altered threshold for TCR activation could

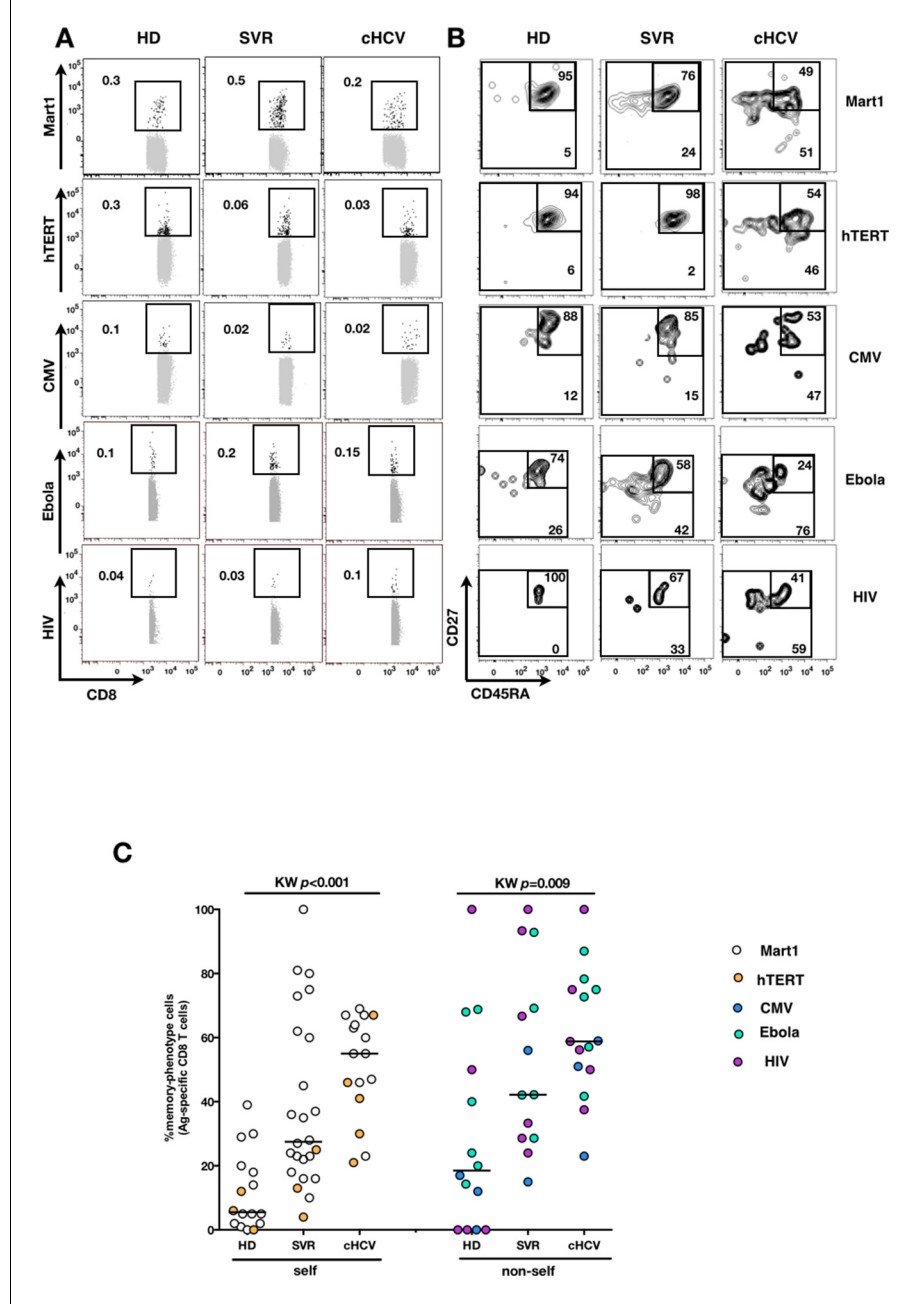

**Figure 3.** Memory-phenotype cells within self and non-self antigen-inexperienced populations. (**A**) Representative examples of Mart1-, hTERT-, CMV-, Ebola- and HIV- specificpopulations from HD, SVR, and cHCV patients. Enriched tetramer-specific populations are overlaid on total CD8$^+$ T cells. (**B**) CD45RA/CD27 phenotype of tetramer-specific populations gated in A. (**C**) Percentages of memory-phenotype cells in Mart1- and hTERT- (self); CMV-, Ebola- and HIV- (non-self) specific populations from HD, SVR and cHCV patients. Further subsetting of MP inexperienced T cells into CD45/CD27-based T cell differentiation phenotype is provided in *Figure 3—figure supplement 1*.

The following figure supplement is available for figure 3:

**Figure supplement 1.** CD45RA/CD27-based subsetting of Mart1-, Ebola-, and HIV- specific T cells enriched from cHCV patients.

---

explain the differentiation phenotype of inexperienced T cells. CD5 expression has been shown to correlate with the threshold of activation in mice (*Grossman and Paul, 2015*). It is typically high on naïve T cells, showing diminished levels as a function of T cell differentiation (*Figure 5—figure supplement 1*). We observed phenotypic changes (i.e. low CD5 expression) that were significant for the

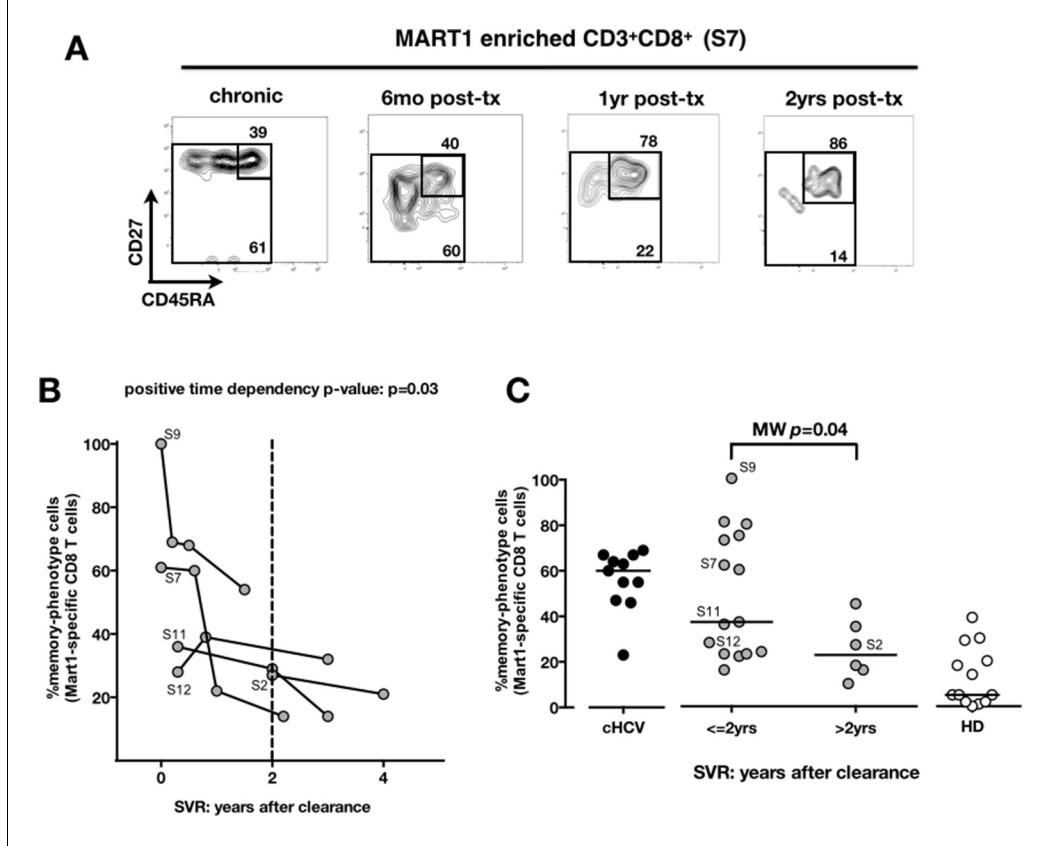

**Figure 4.** Memory phenotype of Mart1-specific CD8+ T cells during chronic infection may be reversed by viral clearance. (**A**) Example of CD45RA/CD27 phenotype of Mart1-specific cells during chronic phase, and over time after viral clearance in one HLA-A0201 SVR patients (patient S7). (**B**) Percentages of Mart1 memory-phenotype cells over time after viral clearance on 5 SVR patients with longitudinal sampling – including S7 presented in E. (**C**) Percentages of memory-phenotype cells in Mart1-specific populations *vs.* time elapsed since clearance of the virus in SVR patients (time-stratified, in years). These data include all HLA $A_{0201}$ SVR patients; first available data is incorporated for follow-up patients presented in F.

comparison between CD45RA+/CD27+ naïve CD8+ T cells in cHCV vs HD (KW p=0.02, *Figure 5A and B*). Based on its role in regulating TCR signaling, we predicted that lower CD5 expression on naïve T cells would result in their hyperactivation upon stimulation. This was tested functionally by evaluating TCR signaling in naïve CD8+ T cells, stimulating cells with low doses of plate-bound anti-CD3 and anti-CD28 Abs. While only weak induction of phosphorylated ERK (p-ERK) could be observed in HD during the first hour of stimulation, TCR stimulation induced a strong p-ERK signal in naïve cells from seven of sixteen cHCV patients tested (histograms from one responding cHCV and one HD are shown in *Figure 5C*; MW p=0.03, *Figure 5D*). Using the same stimulation protocol, we investigated expression of activation markers (*i.e.,* CD25, CD69) measured after 24 hr stimulation. Consistent with p-ERK data, we observed higher percentages of cells expressing CD25 on naïve CD8+ T cells from cHCV as compared to HD (representative example from one cHCV and one HD are shown in *Figure 5E*; MW p=0.02, *Figure 5F*). Similar results were obtained for CD69 analysis (data not shown). We also observed increased percentages of naïve CD8 T cells undergoing activa-tion-induced cell death – as assessed by active caspase 3 staining after 24 hr - in cHCV patients as compared to HD (MW p=0.002; *Figure 5—figure supplement 2*). These findings are all consistent with strong TCR engagement despite the use of low doses of cross-linking antibodies in cHCV patients.

To test the mechanistic link between CD5 expression and hyperactivation of naïve T cells, we evaluated the effect of blocking CD5 signaling. When PBMCs from HD were exposed to blocking anti-CD5 Abs (αCD5) prior to TCR stimulation, we observed (i) increased levels of p-ERK after 5 min (Wilcoxon p=0.007, *Figure 5G*), (ii) increased CD25 expression after 24 hr (Wilcoxon p=0.03,

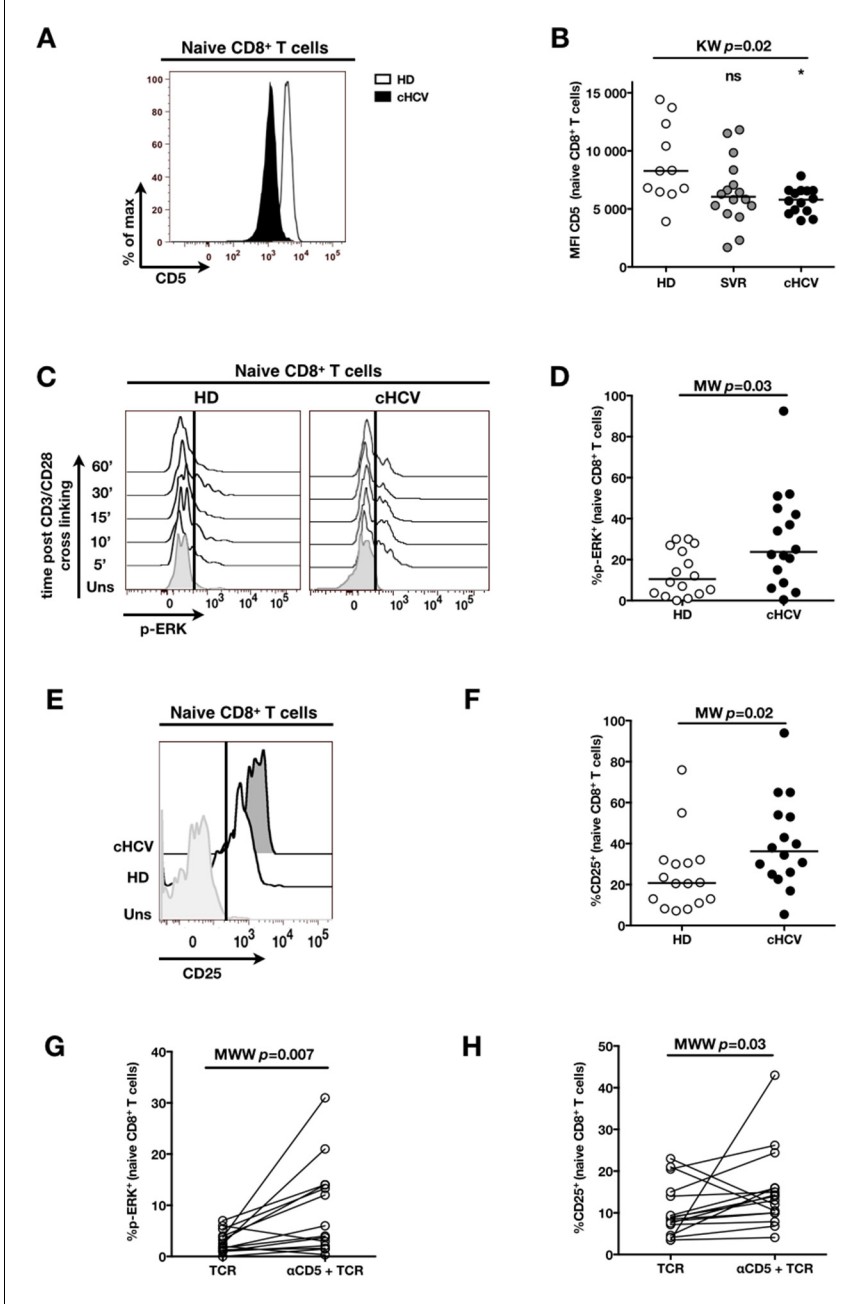

**Figure 5.** Decreased cell surface expression of CD5 on cHCV naïve CD8+ T cells correlates with hypersensitivity to TCR activation. (**A**) Representative histograms of CD5 on naïve CD8+ T cells from one HD and one cHCV patient. (**B**) MFI of CD5 on the surface of naïve CD8+ T cells from HD, SVR, and cHCV patients. Representative histograms and MFI of CD5 on the other T cell differentiation subsets are provided in *Figure 5—figure supplement 1*. (**C**) Representative overlay of histograms of phospho-ERK (p-ERK) signal at different time points following TCR stimulation from one HD and one cHCV patient. Plots are gated on naïve CD8+ T cell populations. (**D**) Percentages of p-ERK positive cells in naïve CD8+ T cells from HD and cHCV patients 5 min after CD3/CD28 stimulation. (**E**) Representative overlay of histograms of CD25 expression, detected at 24 hr after TCR stimulation from one HD, and one cHCV patient. Plots are gated on naïve CD8+ T cell populations. (**F**) Percentages of CD25+ cells in naïve CD8+ T cells from HD and cHCV patients 24 hr after CD3/CD28 stimulation. Representative examples and percentages of active-caspase 3-expressing cells after similar stimulation are provided in *Figure 5—figure supplement 2*. (**G and H**) Percentages of p-ERK (5mins), and CD25 (24 hr) after TCR stimulation in naïve CD8+ T cells from HD, with or without prior CD5 blockade with α-CD5 antibodies. Percentages of active-caspase 3-expressing cells under similar conditions are provided in *Figure 5—figure supplement 3*. Impact of CD5 blockade on TCR activation in cHCV patients is provided in *Figure 5—figure supplement 4*. Similar evaluation of naïve CD8+ T cell repertoire during chronic HBV infection is provided in *Figure 5—figure supplement 5*.

The following figure supplements are available for figure 5:

*Figure 5 continued on next page*

*Figure 5 continued*

**Figure supplement 1.** Evolution of MFI of CD5 over T cell differentiation in HD and cHCV patients.
**Figure supplement 2.** Increased activation-induced cell death after TCR stimulation in cHCV patients.
**Figure supplement 3.** CD5 blockade leads to increased activation-induced cell death after TCR stimulation in HD.
**Figure supplement 4.** Impact of CD5 blockade on TCR activation in cHCV patients.
**Figure supplement 5.** Distinct perturbation of naïve CD8$^+$ T cell repertoire during chronic HBV infection.

*Figure 5H*), and (iii) increased percentages of dying naïve CD8 T cells as assessed by active caspase 3 staining after 24 hr (Wilcoxon p=0.003) (*Figure 5—figure supplement 3*). When compared to cHCV patients, αCD5 partially reproduced the hyperactivation phenotype of naïve T cells from cHCV patients (*Figure 5—figure supplement 4A,B*). By contrast, when αCD5 was applied to the cHCV patients, we observed no further increase in TCR-induced activation (*Figure 5—figure supplement 4 A–D*). These data provide direct evidence for a negative role of CD5 on TCR-induced activation and activation-induced cell death, and support the concept that CD5 molecule is responsible, in part, for the hyperactivation phenotype observed in naive T cells of cHCV patients. Together, these data support a model where low expression of CD5 on naïve T cells in cHCV patients results in dysregulation of the homeostatic TCR threshold.

## Memory phenotype cells can be expanded to generate robust CD8$^+$ T cell responses

We next evaluated the consequences of a low threshold for TCR activation on the ability of inexperienced T cells to expand and differentiate after stimulation with cognate peptide. After 8–11 days of in vitro priming, we observed increased percentages of Mart1-specific CD8$^+$ T cells when expanded from PBMCs of cHCV patients as compared to those from HD (cHCV vs. HD, Day 8, M-W p=0.02, cHCV vs HD, Day 11, M-W p=0.009, *Figure 6A and B*; individual FACS plots for all donors are provided in *Figure 6—figure supplement 1*). The positive impact of chronic infection on naïve T cell expansion was titratable, with more striking differences in the proportion of MP cells after expansion observed when cells were primed with high doses of peptide (*Sprent and Surh, 2011*) (Day 8, M-W p=0.03; *Figure 6—figure supplement 2*). Finally, we found that the Mart1-specific CD8$^+$ T cells generated from cHCV patients express slightly higher amounts of granzyme B (representative example from three cHCV and three HD is shown in *Figure 6C*; MW p=0.02, *Figure 6D*). Interestingly, a tendancy for similar differences in Granzyme B expression could be seen in freshly isolated Mart1-specific CD8$^+$ T cell populations in cHCV patients (M-W p=0.09 as compared to HD, *Figure 6—figure supplement 3*). Hyperreactive preimmune repertoire was further supported by our observation of increased secretion of IFNγ by freshly isolated and antigen-restimulated cells – shown for Mart1, hTERT and CMV peptides in tumor-free, CMV-seronegative cHCV donors (2-way Anova p=0.0002; *Figure 6E and F*).

Together, our results favor a model where low levels of CD5 on naïve-phenotype cells from cHCV donors allow low-affinity interactions with non-cognate antigens to result in T cell differentiation, thereby providing an explanation for the increased frequency of MP cells in cHCV patients. Additionally, our data indicate that qualitative alterations of the CD8$^+$ T cell preimmune repertoire in cHCV patients may result in a boosted response to cognate immune stimulation.

## Distinct preimmune repertoire perturbations during chronic HBV infection

Testing our ability to identify preimmune repertoire perturbations in other clinical conditions, we collected 18 cHBV patients using standard sampling procedures. We found normal percentages of CD3$^+$CD8$^+$ T cells (data not shown), and decreased percentages of bulk naïve CD8$^+$ T cells (MW p=0.009, *Figure 5—figure supplement 5A*). With the limited amount of cells available, we focused our analysis to (i) absolute count and phenotype of Mart1-specific T cells, (ii) CD5 expression on bulk

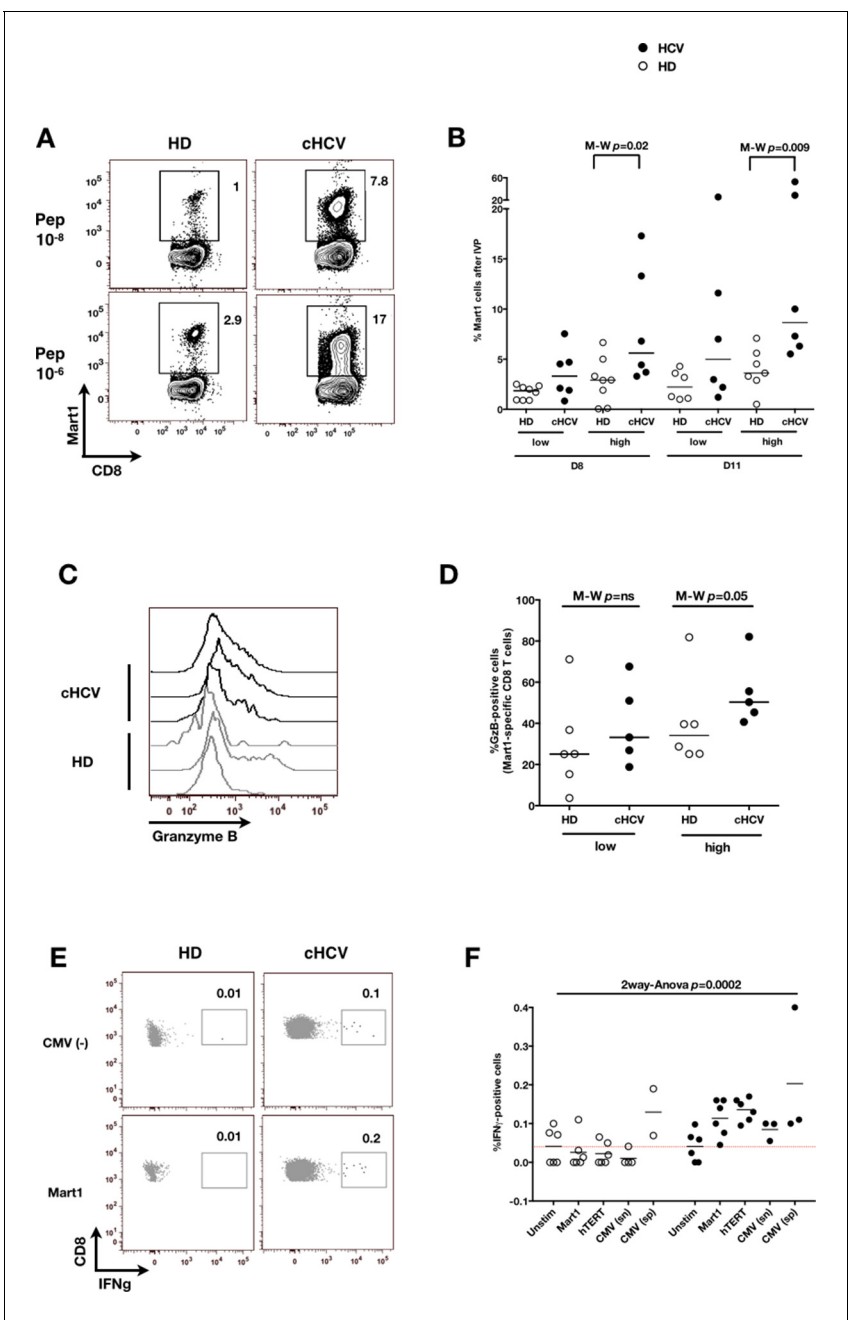

**Figure 6.** Memory phenotype cells can be expanded to generate robust CD8+ T cell responses. (A) Examples of Mart1-specific populations expanded from HD and cHCV patients after 8 days of in vitro priming (IVP) with low (10–8, upper line) and high (10–6, bottom line) doses of Mart1 peptide. FACS plots from all donor tested are provided in **Figure 6—figure supplement 1**. (B) Percentages of Mart1-specific cells expanded after 8 and 11 days of IVP with low and high doses of Mart1 peptide as in A. Proportions of MP cells within those expanded populations are indicated in **Figure 6—figure supplement 2**. (C) Representative histograms of intracellular granzyme-B expression by Mart1-specific T cells expanded from 3 HD and 3 cHCV after 8 days of IVP with high doses of peptide as in A. (D) Percentages of granzyme-B-expressing Mart1-specific T cells expanded from HD and cHCV patients after 8 days of IVP with low and high doses of Mart1 peptide. Baseline percentages are indicated in **Figure 6—figure supplement 3**. (E) Representative examples of IFNγ detection intracellularly after in vitro restimulation with CMV or Mart1 peptides in CMV seronegative, tumor-free HD and HCV patients. IFNγ-positive populations are overlaid on total CD8+ T cells. (F) Percentages of cells with IFNγ-positive staining after Mart1-, hTERT-, and CMV- in vitrorestimulation in HD and cHCV patients. sn, seronegative; sp, seropositive.

The following figure supplements are available for figure 6:

**Figure supplement 1.** FACS plots of Mart1-specific populations expanded in vitro from cHCV and HD.

*Figure 6 continued on next page*

*Figure 6 continued*

**Figure supplement 2.** Increased proportions of memory-phenotype cells within Mart1 populations expanded from cHCV patients.

**Figure supplement 3.** Baseline proportions of granzyme B-expressing cells in cHCV patients and HD.

naïve T cells, and (iii) response to TCR cross-linking. We demonstrated lower absolute numbers of Mart1-specific CD8$^+$ T cells in HBV patients (MW p=0.003, *Figure 5—figure supplement 5B*) and increased frequencies of MP Mart1-specific cells (MW p=0.03, *Figure 5—figure supplement 5C*) as compared to HD, but (ii) similar levels of CD5 expression (MW p=*ns*, *Figure 5—figure supplement 5D*), and (iii) a similar a activation profile of bulk naïve T cells as compared to HD (MW p=*ns*, *Figure 5—figure supplement 5E*). These results indicate that different persistent viral infections of the liver can trigger distinct preimmune repertoire perturbations. Additional studies will be required to fully evaluate the heterogeneous disease pathogenesis of HBV infections as reflected by the observed immune phenotypes.

## Discussion

Our study provides novel evidence for chronic viral infection as a cause of CD8$^+$ T cell preimmune repertoire dysregulation. Specifically, we demonstrated that naïve CD8$^+$ T cells are dysregulated in the context of cHCV, marked by (i) decreased sjTRECs levels, (ii) a restricted Vβ repertoire, and (iii) a lower threshold for TCR engagement.

Prior examples suggestive of preimmune repertoire perturbations have been documented in humans. An increased threshold for TCR activation in naïve CD4$^+$ T cells in elderly persons has been proposed as participating in the diminished response to vaccination that occurs with increasing age (*Li et al., 2012*). Conversely, a decreased threshold for TCR activation, secondary to sustained cytokine production, leads to diverse autoimmune manifestations in rheumatoid arthritis patients (*Singh et al., 2009*; *Deshpande et al., 2013*). With respect to chronic infection, functional defects in the naïve T cell compartment have also been documented in HIV-infected individuals, with non-cognate activation of T cells correlating with disease progression (*Favre et al., 2011*). One major caveat for these studies is that their analysis was limited to global dysregulation of the bulk naïve T cell repertoire.

The challenge of studying perturbations of antigen-specific populations is their low precursor frequency. Taking advantage of the possibility to study viremic *vs.* cured patients, we chose to investigate this question in cHCV patients. Analyzing rare (*i.e.*, frequency = $10^{-7}$ - $10^{-5}$) antigen-specific inexperienced CD8$^+$ T cells populations, we show increased proportions of memory-phenotype cells in cHCV patients, and demonstrate that this correlates with naïve T cells being hyperreactive to TCR signaling in the context of the chronic infection. Despite these altered phenotypes, the absolute number of antigen-specific cells was comparable to healthy donors. Of note, cHCV patients are not thought to experience altered thymic output. As such, our findings provide direct evidence that MP antigen-specific T cells can arise in non-lymphopenic humans.

It has been suggested that a high degree of cross-reactivity with environmental antigens is the trigger for differentiation and MP conversion (*Sprent and Surh, 2011*). This finding has been reported for human viral peptide / MHC restricted CD4$^+$ T cells in unexposed donors (*Su et al., 2013*). While cross-reactivity is a possible explanation for our findings, we demonstrate in cured patients that the antigen-specific inexperienced T cell populations are restored to a naïve phenotype. This result will need to be confirmed in a larger longitudinal cohort study. It favors an alternative model, where homeostatic proliferation accounts for the perturbed naïve T cell repertoire in cHCV patients. Supporting this conclusion, we note the evidence for rapid reversibility to a healthy preimmune repertoire after transient lymphopenia (*Jones et al., 2013*). Consistent with our findings, Jones et al. studied multiple sclerosis patients and showed an anti-CD52 (also known by alemtuzumab) treatment-induced narrowing of the Vβ repertoire and the dilution of sjTREC after treatment, with a complete restoration of normal levels two years post-therapy (*Jones et al., 2013*).

Infection and inflammation is known to lower the threshold of TCR signaling in memory T cells, making them more sensitive to activation (*Richer et al., 2013*). This effect is mediated by

inflammatory cytokines (*Raué et al., 2013*). Our results extend this concept to naïve T cells and introduce CD5 downregulation as a mechanism for hyperreactivity. CD5 tunes the TCR signaling threshold in peripheral T cells, with naïve cells expressing higher levels than central memory or effector T cells (*Tabbekh et al., 2013*). In mice, Hawiger *et al* demonstrated that anti-CD5 blocking antibodies, or the use of CD5$^{-/-}$ transgenic MOG-specific T cells, resulted in higher sensitivity to experimental autoimmune encephalitis (*Hawiger et al., 2004*). In B cells, CD5 has also been shown to regulate activation and low CD5 expression correlates with high sensitivity to activation induced cell death (*Tabbekh et al., 2013*). In line with these findings, we demonstrate an increased sensitivity of CD5$^{lo}$ naïve CD8$^+$ T cells to TCR ligation in cHCV patients. We further provide direct evidence that this hypersensitivity phenotype can be partially reproduced in HD by blocking CD5. While not evaluated in our patient cohort, we propose that elevated levels of inflammatory cytokines may be responsible for the altered CD5 expression on naïve cells (*Park and Rehermann, 2014*). Finally, we applied our strategy for evaluating preimmune repertoire perturbations to other clinical conditions, and demonstrate in cHBV patients that a distinct persistent infection of the liver triggers a different preimmune signature. This observation may be related to the differing innate inflammation induced as a result of infection (*Duffy et al., 2014*).

The combination of low levels of CD5 and increased proportions of MP in inexperienced antigen-specific populations may provide a compounded effect, resulting in a highly reactive CD8$^+$ T cell compartment. We provide evidence here that chronic HCV infection facilitates the generation of robust self-specific responses from the pool of preimmune cells. Given the important role for cellular immunity in the pathogenesis of autoimmune manifestations (*Palermo et al., 2001*), we speculate that circulating self-reactive effector CD8$^+$ T cells may contribute to the systemic immune activation observed during chronic HCV infection, and account for some of the extra-hepatic autoimmune-like manifestations (*Lee et al., 2012*). If our prediction is correct, the ability to restore a physiologically normal preimmune repertoire in cured patients may thus justify early treatment as a means to limit immune-mediated manifestations of the disease. Further investigation in longitudinal cohorts is warrented to confirm these hypotheses, as well as assess the impact on the generation of non-self-specific responses (e.g., in the context of vaccination).

In summary, our study demonstrates that naïve CD8$^+$ T cells are dysregulated during cHCV, with marked perturbations of the preimmune repertoire. Specifically, low levels of CD5 at the surface of naïve T cells, and high proportions of memory-phenotype cells represent two mechanisms by which antigen-inexperienced CD8$^+$ T cells are susceptible to stimulation and antigen-induced expansion. These findings should be considered when designing future immunotherapeutic strategies.

## Materials and methods

### Human subjects, blood samples processing and HLA typing

29 cHCV, 37 SVR, and 18 cHBV patients were included (*Table 1*). All subjects were followed in the Liver Unit of Hôpital Cochin (Paris, France) or the Department of Internal Medicine II (Freiburg, Germany). French samples were obtained as part of study protocol C11-33 approved by the INSERM clinical investigation department with ethical approval from the CPP Ile-de-France II, Paris (ClinicalTrials.gov identifier: n° NCT01534728). German samples were obtained in the University Hospital Freiburg according to regulations of local ethic committee. Both study protocols conformed to the ethical guidelines of the Declaration of Helsinki, and patients provided informed consent. Patient peripheral blood mononuclear cells (PBMCs) were obtained from leukapheresis, or whole blood collections. Healthy donor PBMCs were obtained from buffy coat preparations or whole blood collections (Etablissement Français du Sang, France). PBMCs were processed within 5 hr of their collection. They were used either fresh, or frozen and thawed when needed – and in both cases, cells were rested overnight in serum-free RPMI at 37° before performing functional studies. Absolute lymphocyte counts were determined on the day of collection at the hospital laboratories for HCV and SVR patients, and on fresh samples using AccuCheck Counting Beads (Life Technologies, France) for healthy donors. For all samples, PBMCs were isolated by Ficoll-Paque gradient separation (GE Healthcare, France) after 1:4 dilution in RPMI1640 (Gibco, Life Technologies, France) and controlled for viability (>90%). Molecular HLA-A and –B loci typing were determined using extracted genomic DNA according to standard clinical laboratory procedures (Hôpital St Louis, Paris, France).

## MHC class I multimers

Photocleavable-HLA-A*02:01 multimers were constructed using peptide exchange technology as previously described (*Jenkins et al., 2010*; *Toebes et al., 2006*; *Altman and Davis, 2003*; *Hadrup et al., 2009*). Briefly, heavy chain of HLA-A$_{0201}$ and β2m were produced separately in *E. coli*. Refolding was achieved by diluting each subunit in buffer containing the A$_{0201}$ UV photocleavable peptide (KILGFVFJV, 95% purity, PolyPeptide, France) (*Toebes et al., 2006*; *Blattman et al., 2002*). After biotinylation with recombinant BirA enzyme (Avidity, Denver, USA), monomers were selected by size exclusion chromatography (Akta Purifier, GE Healthcare, France) and stored at -80°C until use. For specific peptides, synthetic 9mer were purchased (75% purity, BioMatik, Toronto, Canada): MART1$_{26-35(Leu27)}$ (ELAGIGILTV), hCMV pp65$_{495-503}$ (NLVPMVATV), hTERT1$_{572-580}$ (RLFFYR-KSV), Ebola NP$_{202-210}$ (RLMRTNFLI)(*Sundar et al., 2007*), and HIV-1 Gag p17$_{77-85}$ (SLYNTVATL). 200 μM peptides were exchanged on calculated amounts of monomers (2 μM final concentration) for 1h under UV-lamp (366nm, 2*8W, Chromacim, France). Titrated amounts of PE or APC-streptavidin (Invitrogen, France) were added. After incubation with D-biotin (25 μM final, Sigma, France), fluorescently labeled multimers were kept in the dark at 4°C until use. Mart1 PE pentamers were purchased (ProImmune, UK) as quality control for our in-house production.

## Tetramer associated magnetic enrichment (TAME) of antigen-specific CD8$^+$ T cells

TAME was performed as previously described (*Alanio et al., 2010*; *Alanio et al., 2013*; *Kyewski and Klein, 2006*). Briefly, purified PBMCs ($2 \times 10^7$ to $4 \times 10^8$) were incubated with FcR blocking reagent (Miltenyi, France), then stained with PE and/or APC pMHC-multimers at 20nM final concentration for 30 min. Samples were incubated with anti-PE-microbeads and positive selection was performed using MS MACS separation columns (Miltenyi, France). Unbound cells ("Depleted" fraction) were collected. Bound cells ("Enriched" fraction) were eluted. As previously published (*Alanio et al., 2010*), tetramer-positive populations were gated as LiveDump$^-$CD8$^+$Tetramer$^+$ cells. To approximate the number of the epitope-specific T cells within each sample, we used a calculation previously described by Moon *et al* (*Arstila et al., 1999*; *Moon et al., 2009*). Precursor frequency is defined as the number of tetramer-positive events in the "Enriched" fraction divided by the number of total CD8$^+$ in the sample.

## Ab staining, flow cytometry and cell sorting

PBMCs were stained with titrated amounts of monoclonal Ab (mAbs) obtained from BD Biosciences, Biolegend, or eBiosciences (*Supplementary file 1*). Live/Dead Fixable Aqua reagent (Life Technologies, France) was included at the same incubation step (dilution 1/200) in order to exclude dead cells. For PhosFlow experiments, cells were stained with surface Abs for 20 mins, then fixed with PFA 3.2% for 10 min at 37°C, and permeabilized by addition of 90% methanol on ice. Intracellular staining of granzyme B was performed using Transcription Factor Buffer Set (BD Biosciences). Samples were acquired using an LSR Fortessa cell analyzer (BD Biosciences, France). Data were analyzed using FACS DIVA 6.0 (BD) and FlowJo 8.8.7 (Tree Star) softwares. Where indicated, stained cells were sorted using a FACS AriaII (BD) in a P2$^+$ facility.

## Intracellular cytokine staining

Human PBMCs were rested overnight in RPMI 1640 GlutaMAX-10% pooled human serum. Cells were plated at $5 \times 10^6$/mL in 24-well plates, and restimulated in vitro with MART1$_{26-35(Leu27)}$, hCMV pp65$_{495-503}$, or hTERT1$_{572-580}$ peptides (10 μM final). After 1 hr of stimulation, GolgiPlug (5 μg/mL final, BD) was added. After 7 hr, cells were stained for surface Abs, then intracellularly using standard procedures (Cytofix/Cytoperm; BD).

## sjTRECs quantification

One million FACS-sorted T cells were lysed in TRIzol Reagent (Life Technologies, France). Genomic DNA was extracted following manufacturer's instructions. Quantification of thymic sjTREC was performed by RT-PCR (ABI PRISM7700; Applied, France) (*Obar et al., 2008*; *Moon et al., 2007*; *Moon et al., 2009*; *Talvensaari et al., 2002*). Data were expressed per 150 000 cells, after normalization for the albumin genomic copy number.

## Immunoscope

After TAME, 1500 naïve and memory Mart1-specific CD8[+] T cells were sorted into RLT Buffer (Qiagen, France). Total RNA was extracted (Qiagen Microkit). cDNA were generated using the Supercript II enzyme (Invitrogen, France). RT-PCR reactions, thermal cycling conditions, calculations for relative usage of each Vβ family, and immunoscope profiles were performed as previously described (*Alanio et al., 2013*; *Bouvier et al., 2011*) (*Supplementary file 2*).

## Determination of naïve Vβ families by flow cytometry

One million PBMCs were stained for T cell surface markers and a set of three Abs directed against TCR-Vβ families (*Supplementary file 3*; IOTest Beta Kit, Beckman/Coulter, France). TCR-Vβ families were classified in increasing order of percentage usage. The Lorenz curve was constructed as a graphical representation of the diversity of the repertoire (*Alanio et al., 2010*; *De Maio, 2007*). After ordering Vβ chains by abundance, from lowest to highest, the Lorenz curve shows the cumulative distribution : for a given percentage (x) of the 24 Vβ chains, it indicates the proportion of the T cell population which have Vβ chains that are among the 24 * x% least abundant ones. Gini coefficient was calculated as the ratio of « area between the line representing equal use of all Vβ chains (dotted line) and the observed Lorenz curve » to « total area below the line representing equal use ». As such, the higher the Gini coefficient, the more unequal the distribution is.

## In vitro TCR activation assays

96-well plates were coated overnight with biotin anti-human CD3 and anti-human CD28 (1 µg/mL and 0.5 µg/mL final concentration, respectively). Unstimulated and PMA/Ionomycin conditions (50 ng/mL and 1 µg/mL respectively) were used as negative and positive controls. Measurements for T cell activation included: PhosFlow, as described above; and phenotypic activation, as measured by expression of CD25 and CD69 following a 24–48 hr culture. For experiments with blocking CD5, cells were preincubated with 5µg/mL anti-human CD5 for 1 hr before being plated for TCR stimulation.

## In vitro priming of antigen-specific CD8[+] T-cell precursors

PBMCs from HLA-A*0201-positive donors were primed in vitro using the ELAGIGILTV (ELA) peptide derived from Melan-A/MART-1 antigen (residues 26–35), using previously published method with minor adaptations (*Martinuzzi et al., 2011*). Briefly, thawed PBMCs were resuspended in AIM medium (Invitrogen), plated at $5 \times 10^6$ cells/well in a 24-well tissue culture plate, and stimulated with 10nM (low dose, 10–8) or 1 µM (high dose, 10–6) of Mart1 peptide ELAGIGILTV in the presence of GM-CSF (0.2 µg/ml, R&D Systems). After 24 hr, dendritic cells maturation was induced by the addition of a cytokine cocktail comprising TNF-α (1000 U/mL), IL-1β (10 ng/mL), IL-7 (0.5 ng/mL) and PGE2 (1 µM) (R&D Systems). On day 2, fetal calf serum (FCS; Gibco) was added to reach 10% by volume per well. Fresh RPMI-1640 (Gibco) enriched with 10% FCS was used to replace the medium every 3 days. Frequency and phenotype of ELA-specific CD8[+] T-cells were determined on day 8–11.

## CMV serology

CMV serology was determined on plasma samples from HD and HCV patients by ELISA for CMV-specific IgG Abs (Liaison XL, Diasorin). Donors were defined as seropositive for CMV if specific IgG>13 U/mL, and seronegative if IgG<13 U/mL.

## Statistical analysis

Statistics were performed using Prism 5, GraphPad software (San Diego, USA). Single continuous variable data were analyzed by Mann-Whitney (MW), or Kruskal-Wallis (KW) followed by Dunn's Multiple Comparison Test. Multi-feature continuous variable data sets were analyzed by Anova and Bonferroni post-test. Paired non-parametric datasets were analysed using Wilcoxon's statistical test. Correlation were analysed using Spearman linear regression. For all these tests, a cut-off value of $p \leq 0.05$ was chosen (*$p \leq 0.05$; **$p \leq 0.01$; ***$p \leq 0.001$). For longitudinal data on SVR patients, after linearisation of the data by squaring, a mixed model was fitted with a fixed time effect and random patient effects for both the slope and the intercept. p-value gives significance for the fixed slope effect. The R function lme (package nlme) was used.

## Acknowledgements

The authors wish to thank all patients for participating in the study. We thank the Etablissement Français du Sang (EFS, France) for healthy donors samples; P. Loiseau and the Laboratory of Immunology (Hôpital Saint Louis, Paris, France) for molecular HLA typing; M. Ahloulay, T. Buivan, K. Sperber, and A. Noble for coordination of the clinical protocol; V. Rouilly, and M. Fontes for careful reading of the manuscript and support for statistical analysis; A. Alanio and N. Dulphy for logistic assistance; and P. Bousso, F. Lemaitre, and L. Rogge for technical help and critical review of the manuscript. We would also like to acknowledge the Centre d'Immunologie Humaine (CIH, Institut Pasteur, Paris, France) for support and technical expertise, as well as technicians from the Laboratory of Virology (Hôpital Trousseau, Paris, France) for CMV serological testing.

## Additional information

### Competing interests

SP: Has received consulting and lecturing fees from Bristol-Myers Squibb, Boehringer Ingelheim, Janssen, Vertex, Gilead, Roche, MSD, Novartis, Abbvie, Sanofi and Glaxo Smith Kline, and grants from Bristol-Myers Squibb, Gilead, Roche and MSD. The other authors declare that no competing interests exist.

### Funding

| Funder | Grant reference number | Author |
| --- | --- | --- |
| Institut National Du Cancer | | Cécile Alanio<br>Philippe Sultanik<br>Brieuc Perot<br>Darragh Duffy<br>Stanislas Pol<br>Vincent Mallet<br>Matthew L Albert |
| European Research Council | SPHINX | Cécile Alanio<br>Marit M van Buuren<br>Estelle Mottez<br>Antoine Toubert<br>Victor Appay<br>Stanislas Pol<br>Vincent Mallet<br>Matthew L Albert |
| Institut Pasteur | | Darragh Duffy<br>Elisabetta Bianchi<br>Annick Lim<br>Emmanuel Clave<br>Aurélie Schnuriger<br>Kerstin Johnsson<br>Antoine Garbarg-Chenon<br>Laurence Bousquet<br>Stanislas Pol<br>Matthew L Albert |

The funders had no role in study design, data collection and interpretation, or the decision to submit the work for publication.

### Author contributions

CA, FN, Conception and design, Acquisition of data, Analysis and interpretation of data, Drafting or revising the article; PS, AS, Acquisition of data, Drafting or revising the article; TF, Acquisition of data, Analysis and interpretation of data, Contributed unpublished essential data or reagents; BP, Acquisition of data, Analysis and interpretation of data; DD, KJ, JB, AT, Analysis and interpretation of data, Drafting or revising the article; EB, AL, EC, Acquisition of data, Analysis and interpretation of data, Drafting or revising the article; MMvanB, TNS, Conception and design, Drafting or revising the article, Contributed unpublished essential data or reagents; AGC, Acquisition of data, Contributed unpublished essential data or reagents; LB, EM, FH, Conception and design, Contributed unpublished essential data or reagents; VA, RT, SP, Conception and design, Analysis and

interpretation of data, Drafting or revising the article; VM, Conception and design, Drafting or revising the article

### Ethics
Clinical trial registration NCT01534728
Human subjects: 29 cHCV, 37 SVR, and 18 cHBV patients were included (Table 1). All subjects were followed in the Liver Unit of Hopital Cochin (Paris, France) or the Department of Internal Medicine II (Freiburg, Germany). French samples were obtained as part of study protocol C11-33 approved by the INSERM clinical investigation department with ethical approval from the CPP Ile-de-France II, Paris (ClinicalTrials.gov identifier: n°NCT01534728). German samples were obtained in the University Hospital Freiburg according to regulations of local ethic committee. Both study protocols conformed to the ethical guidelines of the Declaration of Helsinki, and patients provided informed consent.

## Additional files

### Supplementary files
• Supplementary file 1. HLA A0201-donors included in the study.

• Supplementary file 2. Abs used for flow cytometry experiments.

• Supplementary file 3. Oligonucleotides used for Immunoscope assay.

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
