## [Decision Letter]

[Editors’ note: this article was originally rejected after discussions between the reviewers, but the authors were invited to resubmit after an appeal against the decision.]

Thank you for choosing to send your work entitled "Bystander hyperactivation of preimmune CD8^+^ T cells in chronic HCV patients can be reversed by viral clearance" for consideration at *eLife*. Your full submission has been evaluated by Tadatsugu Taniguchi (Senior editor), a guest Reviewing editor, and two peer reviewers, and the decision was reached after discussions between the reviewers. Based on our discussions and the individual reviews below, we regret to inform you that your work will not be considered further for publication in *eLife*.

The reviewers found the story interesting but they also found that the data was very preliminary. Particular concerns were raised regarding the lack of clear information on the treatment the patients had received and which patients were included in the different experiments figures and the speculative nature of the Discussion with little offered in terms of clinical observations that would match the authors' findings.

A better explanation of the patient cohort and what exactly was done in several of the experiments would improve the manuscript, especially as testing the physiological significance of these changes in the composition of foreign-antigen specific T cells will be difficult to determine in humans.

Testing the physiological significance of these changes in the composition of foreign-antigen specific T cells will be difficult to determine in humans, but we would be forgiving of this limitation if the manuscript was more clear-cut. Given the varied major concerns listed by the reviewers, this report would need to be radically improved in order for the authors to make definitive conclusions.

Reviewer #1:

Alanio et al. have analyzed tumor – and virus-specific cells in tumor – and virus negative subjects and found differences for these populations between healthy subjects, people with chronic and treated HCV infection and those with chronic HBV infection. They found specific T cells with different phenotypes in these populations, notably significant numbers of memory T cells in addition to naïve cells in chronic HCV. They suggest this is due to lower CD5 expression and thus more cell death in naïve cells in chronic HCV.

Overall I think there are quite a few intriguing observations in this study, but at the same time I do have a few questions for the authors and also a few concerns about some of the data. Currently it is not easy to follow the interpretations of the study and to see the importance of the observations. Specifically:

1) It is not fully clear to me that the authors have a strong case that the distinct phenotype they observe has also actual clinical consequences. They discuss autoimmunity and also that antigen-experienced cells in HCV infection might be more prone to stimulation and cell death. But this is all rather vague and speculative. One could equally argue that the number of truly naïve cells remained actually stable and that just some additional memory-like cells were observed, with unknown consequences.

2) The paper does not detail what kind of therapy the patients were receiving (and almost all of them were already treatment experienced in both the chronic and the SVR group). Assuming they had received IFN before, could that not have had significant impact on the phenotype of the T cells? And what was the latest therapy in the SVR group? IFN versus DAA would make an important difference. Also, cells from the healthy subjects were at least partially obtained from buffy coats. Depending on how those are obtained (leftovers from the blood bank?) this could also influence both cell numbers and phenotype. Furthermore, nothing is known about age and other clinical parameters for HD, yet those could have significant impact on results (often HD tend to be younger).

3) The authors also observed decreased number of naïve cells in the case of HBV infection, but none of the other findings extended to this patient group. This is curious and warrants at least some discussion.

4) I am somewhat concerned about the approach to define naïve cells by CD45RA and CD27 alone, based on the data in Figure 1—figure supplement 1. I would argue that this figure shows a significant proportion of this population to be CD127 negative, CD45RO positive and CCR7 negative. The latter two markers are also surprisingly hard to evaluate, since the staining of the total CD8 population indicates inadequate separation between positive and negative cells.

5) The data in Figure 3 needs more detail. In Figure 3, was the treatment IFN? And on how many specific cells are these plots based? They look not like the best flow data. In Figure 2, which patients were studied at which time-point? The range for naïve cells in chronic HCV ranges from 25% to 90%, so the selection of patients in the different post treatment timeframes could greatly alter the results, given the small numbers of data points. Did all subjects show an increase in naïve cells from chronic to post treatment?

6) For some of the analyses it would be important to know how many specific cells were actually included. For example the ICS data (Figure 4) and the immunoscope study (Figure 4) could be greatly influenced/skewed by low cell numbers in the analysis. For the ICS experiment, flow plots should be provided as well, and the results not be reported as a multiple of unstimulated versus unstimulated cells as this not allowing a meaningful interpretation of the data.

7) I think the data in Figure 6 needs to be analyzed differently. The focus on fold change is too dependent on the unstimulated background, which seems higher for the HD sample. The different timpanist also seems to be quite heterogeneous, yet only one time-point is used. I find these data not fully convincing. Were buffy coat cells used for HD? And if so, how long after blood draw were the cells collected?

8) Can the data in Figure 7 really be interpreted? The variation in that assay seems to be extremely high. At least it would be good to show the results as individual data points instead of bar graphs.

Reviewer #2:

The manuscript by Alanio et al. examines memory phenotype CD8 T cells in the preimmune repertoire from chronically infected HCV patients compared to HCV-cured patients and healthy donors. The authors found an increased proportion of pre-immune CD8^+^ T cells exhibited a memory phenotype (rather than the expected naïve phenotype) in chronically infected HCV patients. This correlated with decreased CD5 expression, increased stimulation following TCR stimulation and increased caspase 3 cleavage following activation, suggesting increased sensitivity to TCR stimulation and activation-induced cell death. In vitro mechanistic studies showed that blocking CD5 on CD8 T cells from healthy donors recapitulated aspects of the increased sensitivity to TCR stimulation phenotype seen for CD8 T cells from chronic HCV patients. Furthermore, the authors showed that naïve CD8 T cells from chronic HCV patients had decreased TREC levels and a more restricted Vβ repertoire, indicative of a perturbed naïve CD8 T cell repertoire in chronic HCV patients. Overall, the study has been well designed and the findings add important information to our knowledge of memory phenotype CD8 T cells in humans in the context of chronic viral infection. However, there are some issues with the manuscript that are detailed below.

1) Of the three antigen-specific populations studied, two were specific for self antigens – MART1 and hTERT. Thus, although the authors state that these T cells are antigen "inexperienced", the antigen is present in normal tissues and it seems feasible that chronic HCV infection leads to enhanced presentation of peptides from these self antigens, and subsequent antigen-specific T cell stimulation. In which case the acquisition of non-naïve phenotype is not a surprise. These concerns are partly offset by additional analysis of hCMV specific CD8^+^ T cells (Figure 4), since the frequency of this population is the lowest of all the specificities studied (Figure 4), making it more difficult to be confident about apparent phenotypic differences. It would be useful if the authors could extend their observations to another non-self antigen (ideally, one with a higher precursor frequency), but at the very least the fact that most of the study is focused on self-specific T cells should be discussed in more depth.

2) The data on TCR sensitivity and CD5 expression is interesting, and would be in line with the authors' suggestion that proliferation of naïve phenotype CD8^+^ T cells (as evidenced by loss of TREC – Figure 2) might lead to decreased CD5 expression levels. However, the data shown in Figure 7 do not make a strong case that CD5 expression levels are the root cause of the increased TCR sensitivity of the naïve cells from cHCV patients. The experiments presented show that blockade of CD5 leads to an enhanced response by CD8^+^ T cells – in line with many previous studies. What is needed is to test whether this blockade would normalize the differences in sensitivity between T cells from HD and cHCV donors. To be specific, the authors need to apply anti-CD5 blockade to *both* HD and cHCV T cells and determine whether TCR hypersensitivity is now equivalent between the groups (supporting the authors' idea that CD5 expression levels are the key determinant). Alternatively, they may find that T cells from cHCV patients still show the same degree of enhanced reactivity following TCR engagement – a result that would imply that CD5 blockade increases sensitivity by both populations to the same extent, and that other features of the cells from cHCV patients are responsible for their enhanced sensitivity.

3) The authors show that the proportion of antigen-specific CD45RA^+^CD27^+^ naïve CD8 T cells is reduced in chronic HCV patients compared to healthy donors and HCV-cured patients. The stated conclusion is that chronic HCV infection results in increased memory phenotype CD8 T cells; however, it would be useful for the authors to define the phenotypic traits of the non-naïve populations and show whether there is a consistent increase in either CD45RA^-^CD27^+^ or CD45RA^-^CD27^-^ antigen specific CD8^+^ T cells in chronic HCV patients compared to healthy donors and HCV-cured patients. There appears to be substantial differences between groups or experiments in the flow cytometry plots – e.g. comparing the CD27 expression by non-naïve MART-specific cells in Figure 3 and Figure 3 – and it would be good to know whether this averages out with compiling samples from multiple individuals.

[Editors’ note: what now follows is the decision letter after the authors submitted for further consideration.]

Thank you for resubmitting your work entitled "Bystander hyperactivation of preimmune CD8^+^ T cells in chronic HCV patients" for further consideration at *eLife*. Your revised article has been favorably evaluated by Tadatsugu Taniguchi (Senior editor), a guest Reviewing editor, and two reviewers. The manuscript has been improved but there are some remaining issues that need to be addressed before acceptance, as outlined below:

In general, both reviewers are satisfied with the revised version. Reviewer #1 has raised some issues that still need to be addressed before your manuscript can be accepted. However, no additional experiments are required. You just need to address these concerns by making changes in the text. Regarding Figure 11, please include these data in the manuscript as a supplemental figure.

Reviewer #1:

Alanio et al. have significantly revised their manuscript on bystander activation of unprimed CD8 T cells in chronic HCV infection. I appreciate the amount of work, especially given the challenges and limitations when working on human samples. This has definitely improved the manuscript, but, at the same time, I still have a few questions for the authors:

1) The observation that there are changes in the composition of the CD8 T cell pool based on memory markers remains intriguing. I am still not sure, however, whether this is a relative or an absolute change. The reasons are that the gates on the naïve populations are different for each subject (as seen in Figure 1), making the quantification of the relative number of naïve T cells seem a little random. At the same time (if I understood the new information about absolute T cell counts correctly), absolute T cell counts were performed using different methods in HD vs SVR and cHCV patients, raising the possibility that the differences in absolute CD3 numbers as shown in Figure 1—figure supplement 2 could be a result of different methodology. These issues should be reconciled.

2) I find it very difficult to interpret the TCR data. The cross sectional results on bulk naïve cells in Figure 1 show somewhat subtle differences that could be caused by a multitude of factors. Data on specific cells like from the one subject in Figure 2 should be more revealing. In this case, it seems clear that just one clonotype (already dominant in the naïve cells) is responsible for all memory cells. Would that not imply that one clonotype might have some cross-reactivity and thus have expanded?

3) The analysis of specific cells for naïve versus memory population has been significantly strengthened by the new Figure 3, justifying the overall conclusion that more T cells display a memory phenotype. I have still some concerns, especially regarding the longitudinal and cross-sectional analysis of MART responses (Figure 2 and Figure 4), given that the gates on naïve cells seem to be different in every plot. How was the "right" gate determined in each case? Where the different time-points for a patient not stained in a single experiment or why do the staining patterns/signal strength look so different? Regarding the cross sectional analysis in Figure 4, my concern remains that the differences seen in the few patients more than 2 years after treatment (already barely significant) might be driven by patient selection (exemplified by patients S11 and S12 with low memory percentages already pre and post treatment, potentially skewing the results for the 6 subjects included at more than 2 years post treatment). The case that treatment normalizes the phenotype remains not very strong in my opinion (did the bulk CD8 T cells show changes? more data points might be available).

4) The functional experiments using CD5 blockade remain inconclusive, given that an effect is only seen in HD. As this effect is modest and does not lead to the same level of reactivity as seen in the cHCV patients (R6) it seems unclear whether CD5 is the main mechanism here.

5) The in vitro expansion and ICS experiments in Figure 6 are a bit problematic. First, for each of the subjects the naïve MART precursor frequency should be given, since some of the cHCV subjects had relatively high frequencies as seen in Figure 2—figure supplement 2. In any case, assuming specific cells in cHCV expand better, could this not be explained by them being memory and not naïve cells? For the functional data, the difference in granzyme B is modest and no difference is observed directly ex-vivo (Figure 6—figure supplement 3). As for the IFN assay, are the authors really suggesting that populations requiring TAME for visualization can be detected via ICS directly ex-vivo? Detecting such small populations via ex-vivo ICS from 5 million PBMC seems unlikely.

In summary, to me this revision has significantly strengthened the basic observations supporting the notion that HCV infection indeed impacts other unrelated T cell populations. I remain less convinced about the suggested mechanisms behind these findings and also whether HCV treatment really restores the T cell compartment.

Reviewer #2:

The authors have made numerous revisions to the manuscript, which address the major concerns I raised. The only request would be to include the data shown in Figure 11 in the manuscript. These data make, in my opinion, an important point about the impact and selectivity of the anti-CD5 blockade, and offers material support to the authors' hypothesis. This figure could be included in the supplementary material.

---

## [Author Response]

[Editors’ note: the author responses to the first round of peer review follow.]

We have addressed all of the points raised – including the clarification of patient treatment information and the coding of which patients were used in the different studies. It is also much appreciated that the editor and referees acknowledge the inherent limitations for human investigations; accordingly, we have modified the Discussion so that it is more clear what aspects of our predictions are speculative in nature. In light of our new results and the reorganization of the report, we believe that the revised manuscript is considerably improved.

Reviewer #1:

*1) It is not fully clear to me that the authors have a strong case that the distinct phenotype they observe has also actual clinical consequences. They discuss autoimmunity and also that antigen-experienced cells in HCV infection might be more prone to stimulation and cell death. But this is all rather vague and speculative. One could equally argue that the number of truly naïve cells remained actually stable and that just some additional memory-like cells were observed, with unknown consequences.*

We thank the reviewer for pushing us to clarify our message. We have modified the Discussion in order to make the point that the proposed link to the extra-hepatic manifestations observed in cHCV patients is speculative and will need to be evaluated in longitudinal cohort studies. If our prediction is correct, the ability to restore a physiologically normal preimmune repertoire in cHCV patients may justify early treatment as a means to limit immune-mediated manifestations of the disease. We have also performed new experiments that reinforce our prior observations. With respect to antigen-specific inexperienced T cells being more prone to activation, this has been directly tested using ex vivo expansion as readout. We found MART-1 specific T cells from cHCV patients more abundant and possessing greater effector function (i.e., Granzyme B expression) after cognate peptide stimulation, comparing with healthy controls (see revised Figure 6). These findings indicate that qualitative alterations of the CD8 T cell preimmune repertoire in cHCV patients do indeed impact critical aspects of induced immune responses. It also reinforces our conclusion that lower threshold for TCR activation on naïve T cells might trigger antigen-independent T cell differentiation in cHCV patients. As discussed below, our initial results were confirmed when testing virus-specific T cells, reactive to viral antigens for which the donors had not been exposed (e.g., HIV, Ebola).

Regarding the query about absolute cell numbers, the reviewer is correct in their interpretation, and we now have clarified this point in the revised manuscript – the precursor frequency of naïve-phenotype Mart1-specific T cells is unaffected (see Figure 2—figure supplement 2). This result, however, does not alter our conclusions regarding memory-phenotype cells.

*2) The paper does not detail what kind of therapy the patients were receiving (and almost all of them were already treatment experienced in both the chronic and the SVR group). Assuming they had received IFN before, could that not have had significant impact on the phenotype of the T cells? And what was the latest therapy in the SVR group? IFN versus DAA would make an important difference.*

We apologize for the lack of clarity. 62% of our chronic HCV patients were treatment-experienced, of which 2/3 received bitherapy with pegylated interferon (IFN) and ribavirin (RBV), and 1/3 were treated with a combination of a direct antiviral agent (DAA) and interferon. To assess the potential interference, we stratified our patient groups, and as shown in Figure 7, the observed effect of higher percentages of memory-phenotype Mart1-specific CD8^+^ T cells can be seen in both non-treated and previously treated cHCV patients. We have also provided an update to Table 1, indicating that the median delay post-treatment of included cHCV patients was 3.8 years (IQR1-3 3.4-4.2). Given the timing of our collections, we do not believe there to be a risk of prior therapy confounding our results.

Author response image 1.In both cHCV (left) and SVR (right), proportions of memory-phenotype (MP) cells within Mart1-specific populations are not a function of previous drug regimen.**DOI:**
http://dx.doi.org/10.7554/eLife.07916.028

With respect to SVR patients, all patients received prior treatment (by definition): 70% received an IFN/RBV therapy, and 30% were treated with a DAA-based therapy. While the stratification of patients did not indicate a significant difference between persons receiving the different drug regimen, larger cohorts with precise post-treatment collection times would be required to establish the optimal regimen to maximize immunological restoration. We have amended the manuscript, including details about prior treatment regimens.

With the aim of increasing clarity of our manuscript, we also provide a new table ([Supplementary-material SD1-data]) that is stratified for HLA-A_0201_ positive donors, the subset of patients used for studying antigenspecific T cell phenotypes.

*Also, cells from the healthy subjects were at least partially obtained from buffy coats. Depending on how those are obtained (leftovers from the blood bank?) this could also influence both cell numbers and phenotype. Furthermore, nothing is known about age and other clinical parameters for HD, yet those could have significant impact on results (often HD tend to be younger).*

In our study, buffy coats from HD were collected by the French blood bank, according to blood transfusion procedures. PBMCs were processed within 5 hours of their collection. They were used fresh, frozen and thawed when needed. In both cases, cells were rested overnight in serum-free RPMI at 37° before performing functional studies. We systematically controlled sample quality and excluded any samples containing >10% of dying cells. For cytometric studies, we further gated residual dead cell using live/dead probes. Importantly, allowing appropriate comparison, leukapheresis collection from cHCV and SVR patients followed the exact same procedure (e.g., location, transport time, quality control). Additional information is now added to the revised Methods section. Finally, the age of HD is now indicated. Of note, age-matched controls were selected for our study (see Table 1).

*3) The authors also observed decreased number of naïve cells in the case of HBV infection, but none of the other findings extended to this patient group. This is curious and warrants at least some discussion.*

We apologize for the lack of clarity. To address the question of the pre-immune repertoire in HCV patients, we designed a specific protocol for leukapheresis collection, permitting recovery of up to 1010 PBMCs from each donor. This allowed assay development and optimization, and multiple assays to be performed on same donors. Following our initial discoveries in cHCV patients, we were prompted to evaluate our findings in an alternative viral hepatitis disease state. We were fortunate to have access to a bio-repository of HBV samples that permitted rapid testing, however samples were indeed limited, with ~5x107 PBMCs being available per patient. We therefore restricted our analysis to: (i) Mart1-specific T cell phenotype and numbers, (ii) CD5 levels on bulk naïve T cells, and (iii) CD25 and active caspase 3 percentages after 24 hours of activation with CD3/CD28 antibodies, parameters that we identified in the cHCV cohort as the best biomarkers of a perturbed T cell repertoire. As discussed, we demonstrated lower absolute numbers and intermediate proportions of memoryphenotype Mart1-specific cells in cHBV patients, and a similar CD5 expression and T cell activation as compared to HDs. Based on these findings, we believe that different persistent viral infections can trigger distinct T cell repertoire perturbations. We have taken the advice of the reviewer to include all data from cHBV patients in a separate supplementary figure. These results will be followed up in future studies.

*4) I am somewhat concerned about the approach to define naïve cells by CD45RA and CD27 alone, based on the data in Figure 1—figure supplement 1. I would argue that this figure shows a significant proportion of this population to be CD127 negative, CD45RO positive and CCR7 negative. The latter two markers are also surprisingly hard to evaluate, since the staining of the total CD8 population indicates inadequate separation between positive and negative cells.*

We thank the reviewer for the careful analysis of our data, and acknowledge that the gating strategy is crucial. Regarding the gating on naïve T cells based on high levels of CD45RA and CD27, we validated our strategy using additional healthy donors. As shown in Figure 8, CD45RA^+^CD27^+^ gated naïve CD4^+^ and CD8^+^ T cells express the highest levels of CCR7 and lowest level of HLA-DR markers. We also demonstrate a high degree of correlation between percentages of CD45RA^+^CD27^+^ and CD45RA^+^CCR7^+^ naïve CD8^+^ T cells. These data were confirmed in cHCV patients, with >95% bulk and Mart1- specific CD45RA^+^CD27^+^ CD8^+^ T cells also expressing CCR7. We now provide these data in the manuscript as a supplementary figure (Figure S1B). Regarding the concern related to Figure S1, we confirm that there is a minor population (9%) of CD127 negative cells in the displayed CD45RA^+^CD27^+^ gate. These cells are >99% CD45RO negative and CCR7 positive, which we believe allows us to be confident that they are indeed naïve T cells. Interestingly, they express lower levels of CD27 as compared to other naïve cells. Our interpretation is that this subpopulation corresponds to activated naïve T cells that have shed surface CD27, as previously described in the context of graft-versus-host disease (Poulin, 2003).

Author response image 2.Evaluation of CD45RA/CD27 gating strategy.CD45RA^+^CD27^+^ naïve CD4^+^ and CD8^+^ T cells express the highest levels of CCR7 and lowest level of HLA-DR markers (**A**). Also, there is a close correlation between CD45RA^+^CD27^+^ and CD45RA^+^CCR7^+^- gated naïve CD8^+^ T cells (**B**).**DOI:**
http://dx.doi.org/10.7554/eLife.07916.029

*5) The data in Figure 3 needs more detail. In Figure 3, was the treatment IFN? And on how many specific cells are these plots based? They look not like the best flow data.*

We have revised the indicated figure, now revised Figure 4. Specifically, we (i) provide new FACS plots showing the evolution of Mart1 phenotype in one SVR patient collected during chronic phase, 6 months, 1 year and 2 years after successful IFNRBV therapy; (ii) included 4 new HLA-A0201 SVR patients that were sampled at different time points after viral clearance; and (iii) employed statistical methods to analyze longitudinal datasets (p-value represents the significance of the fixed effect of time after linearisation of the data followed by fitting of a mixed model (see Materials and methods for details). Our data confirm on a per patient basis that viral clearance results in the restoration of an expected naïve phenotype for Mart1-specific T cells. Concerning the number of cells, we agree with the reviewer that low number of Ag-specific inexperienced T cells is a challenge. As an example, in Figure 4, we gated a mean of 97 Mart1- specific T cells (range 48-126). As a general strategy, in line with recognized groups in the field (Yu et al., 2015), we interpret a tetramer-gated population as real if containing more than 20 antigen-specific enriched cells. We chose this number based on our initial evaluation of the limit of detection of our assay (please refer to Alanio et al., 2010), established using spiked numbers of CTL clones.

*In Figure 2, which patients were studied at which time-point?*

The manuscript has been modified accordingly, tagging corresponding data in Figure 4 with patient ID. To minimize bias, we utilized data from the first available time point when performing comparisons among the different patient groups (as shown in revised Figure 4).

*The range for naïve cells in chronic HCV ranges from 25% to 90%, so the selection of patients in the different post treatment timeframes could greatly alter the results, given the small numbers of data points. Did all subjects show an increase in naïve cells from chronic to post treatment?*

We evaluated the impact of previous treatment on our observations (see above, Point #2). Additionally, we show here a plot of Mart-1 memory phenotype cells as a function of time-post treatment for cHCV. Aside from the outlier patient with 25% frequency, we observe a narrow range of differentiated cells.

Author response image 3.Percentages of MP cells within Mart1-specific population is not a function of delay post-treatment in cHCV patients.**DOI:**
http://dx.doi.org/10.7554/eLife.07916.030

*6) For some of the analyses it would be important to know how many specific cells were actually included. For example the ICS data (Figure 4) and the immunoscope study (Figure 4) could be greatly influenced/skewed by low cell numbers in the analysis.*

We apologize for the oversight and have amended the manuscript accordingly. As now indicated in the revised manuscript, we performed the immunoscope using 1,500 sorted naïve and memory Mart1-specific T cells. Although we agree with the reviewer that low number of cells could account for restricted repertoire diversity, we believe it is still valuable for repertoire comparison.

*For the ICS experiment, flow plots should be provided as well, and the results not be reported as a multiple of unstimulated versus unstimulated cells as this not allowing a meaningful interpretation of the data.*

For ICS experiments, we stimulated 107 cells in 24-well format plates with cognate peptide. Based on precursor frequencies, we expect ~2-20 inexperienced antigen-specific cells to be available in a given well for peptide re-stimulation. We made the recommended modification of the figure and added additional patients to the study (see revised Figure 6). The new results confirmed our initial observations.

*7) I think the data in Figure 6 needs to be analyzed differently. The focus on fold change is too dependent on the unstimulated background, which seems higher for the HD sample. The different timpanist also seems to be quite heterogeneous, yet only one time-point is used. I find these data not fully convincing.*

We have revised the data analysis and figure accordingly (now revised Figure 5). To extend our initial observations, we included an additional 6 cHCV patients and 6 HD. Of note, there were no significant differences in the background in HD and cHCV patients in these experiments (data not shown).

We have also evaluated the kinetics of p-ERK and CD25 upregulation. These data are provided in Figure 10. Based on these initial observations, we chose 5 min for measurement of p-ERK and 24 hour for monitoring CD25 expression.

Author response image 4.Kinetics of P-ERK and CD25 in naïve T cells from HD and cHCV patients after CD3/CD28 stimulation.**DOI:**
http://dx.doi.org/10.7554/eLife.07916.031

*Were buffy coat cells used for HD? And if so, how long after blood draw were the cells collected?*

As discussed above (see Point #2), functional studies on HD were performed using buffy coats from the blood bank. PBMCs were processed within 5 hours of collection.

*8) Can the data in Figure 7 really be interpreted? The variation in that assay seems to be extremely high. At least it would be good to show the results as individual data points instead of bar graphs.*

To address the raised concerns, we have increased the number of donors tested. Data was analyzed using paired non-parametric methods, confirming a positive effect of CD5 blockade on T cell activation (please see Figure 6).

Reviewer #2:

*1) Of the three antigen-specific populations studied, two were specific for self antigens – MART1 and hTERT. Thus, although the authors state that these T cells are antigen "inexperienced", the antigen is present in normal tissues and it seems feasible that chronic HCV infection leads to enhanced presentation of peptides from these self antigens, and subsequent antigen-specific T cell stimulation. In which case the acquisition of non-naïve phenotype is not a surprise. These concerns are partly offset by additional analysis of hCMV specific CD8^+^ T cells (Figure 4), since the frequency of this population is the lowest of all the specificities studied (Figure 4), making it more difficult to be confident about apparent phenotypic differences. It would be useful if the authors could extend their observations to another non-self antigen (ideally, one with a higher precursor frequency), but at the very least the fact that most of the study is focused on self-specific T cells should be discussed in more depth.*

We thank the reviewer for his/her interest in our study and for the careful analysis of the manuscript. Although we cannot formally rule out the possibility that cells-specific for self-peptides were activated by their cognate antigen, we believe the extension of our observations to T cells specific for non-self (foreign) antigens supports our conclusion of antigen-independent differentiation of preimmune repertoire in cHCV patients. Indeed, our new experiments measuring HIV and Ebola virus epitopes in cHCV patients – all HIV and presumably Ebola seronegative – considerably strengthens our argument (see revised Figure 3).

We have also included new functional data to assess the impact of preimmune alterations. Notably, Mart1 peptide based expansion was higher in cHCV than normal donors, and the responding T cells showed higher levels of effector molecules (see Figure 6 in revised manuscript). Future studies will address how these findings impact the in vivo capacity of cHCV patients to mount adequate CD8 T cell responses – providing direct relevance for optimization of therapeutic vaccination in this patient population. Notably, these data may be relevant in other chronic disease areas.

*2) The data on TCR sensitivity and CD5 expression is interesting, and would be in line with the authors' suggestion that proliferation of naïve phenotype CD8^+^ T cells (as evidenced by loss of TREC – Figure 2) might lead to decreased CD5 expression levels. However, the data shown in Figure 7 do not make a strong case that CD5 expression levels are the root cause of the increased TCR sensitivity of the naïve cells from cHCV patients. The experiments presented show that blockade of CD5 leads to an enhanced response by CD8^+^ T cells – in line with many previous studies. What is needed is to test whether this blockade would normalize the differences in sensitivity between T cells from HD and cHCV donors. To be specific, the authors need to apply anti-CD5 blockade to* both *HD and cHCV T cells and determine whether TCR hypersensitivity is now equivalent between the groups (supporting the authors' idea that CD5 expression levels are the key determinant). Alternatively, they may find that T cells from cHCV patients still show the same degree of enhanced reactivity following TCR engagement – a result that would imply that CD5 blockade increases sensitivity by both populations to the same extent, and that other features of the cells from cHCV patients are responsible for their enhanced sensitivity.*

We thank the reviewer for the positive comments and suggestions. To address the inter-individual variation, we have increased the number of donor tested in TCR activation experiments (see revised Figure 5). As recommended, we also tested the impact of pre-incubation with blocking CD5 antibody on both HD and cHCV groups. As shown in Figure 11, we demonstrate a consistent positive impact of αCD5 on TCRinduced activation, with a corresponding increase in activation-induced cell death. By contrast, cHCV patients do not show evidence of enhanced T cell activation following CD5 blockade. These data support our interpretation that T cells from cHCV patients are functionally perturbed due to their lower expression of CD5, and that further inhibition does not provide additional hyper-activation.

Author response image 5.Impact of CD5 blockade on TCR activation profile in cHCV patients.(**A and B**) Impact of preincubation with anti-CD5 antibodies on% CD25 and% active-caspase 3 after CD3/28 stimulation in cHCV patients. (**C and D**) Summary of impact of anti-CD5 on HD and cHCV patients.**DOI:**
http://dx.doi.org/10.7554/eLife.07916.032

*3) The authors show that the proportion of antigen-specific CD45RA+CD27+ naïve CD8 T cells is reduced in chronic HCV patients compared to healthy donors and HCV-cured patients.*

We have examined the phenotype of non-naïve Mart1-, Ebola- and HIV-specific CD8 T cells. Following the suggestions made by the reviewer, we have observed an interesting pattern of expression leading us to conclude that the state of differentiation for the observed MP cells is closest to that of central memory T cells. We have included this analysis as Figure 3—figure supplement 1 in our manuscript, and added a sentence in the Results section.

*The stated conclusion is that chronic HCV infection results in increased memory phenotype CD8 T cells; however, it would be useful for the authors to define the phenotypic traits of the non-naïve populations and show whether there is a consistent increase in either CD45RA-CD27+ or CD45RA-CD27- antigen specific CD8^+^ T cells in chronic HCV patients compared to healthy donors and HCV-cured patients. There appears to be substantial differences between groups or experiments in the flow cytometry plots – e.g. comparing the CD27 expression by non-naïve MART-specific cells in Figure 3 and Figure 3 – and it would be good to know whether this averages out with compiling samples from multiple individuals.*

The reviewer is correct that the MFI of CD27 is lower in cHCV patients as compared to HD. This is the case for both naïve and MP Ag-specific populations (see Figure 12). As mentioned above (see discussion in reply to reviewer #1 – point 4), a soluble form of CD27 is released after TCR engagement (Hintzen et al., 1991). We believe this might represent an additional feature of naïve cells that have a hyper-activated phenotype. As we did not fully explore this phenotype in our study, we have not included the data.

Author response image 6.Phenotype of Ag-specific inexperienced T cells in cHCV patients.Percentages of naïve, CM, EM, and EMRA – phenotype Ag-specific Mart1-, Ebola-, and HIVspecific T cells enriched from chronic HCV patients.**DOI:**
http://dx.doi.org/10.7554/eLife.07916.033

References:

Hadrup, S.R., Bakker, A.H., Shu, C.J., Andersen, R.S., van Veluw, J., Hombrink, P., Castermans, E., Thor Straten, P., Blank, C., Haanen, J.B., et al. (2009). Parallel detection of antigen-specific T-cell responses by multidimensional encoding of MHC multimers. Nat Meth 6, 520–526.

Hasan, M., Beitz, B., Rouilly, V., Libri, V., Urrutia, A., Duffy, D., Cassard, L., Di Santo, J.P., Mottez, E., Quintana-Murci, L., et al. (2015). Semi-automated and standardized cytometric procedures for multi-panel and multi-parametric whole blood immunophenotyping. Clin Immunol 157, 261–276.

Hintzen, R.Q., de Jong, R., Hack, C.E., Chamuleau, M., de Vries, E.F., Berge, ten, I.J., Borst, J., and van Lier, R.A. (1991). A soluble form of the human T cell differentiation antigen CD27 is released after triggering of the TCR/CD3 complex. J Immunol 147, 29–35.

Poulin, J.F. (2003). Evidence for adequate thymic function but impaired naïve T-cell survival following allogeneic hematopoietic stem cell transplantation in the absence of chronic graft-versushost disease. Blood 102, 4600–4607.

Yu, W., Jiang, N., Ebert, P.J.R., Kidd, B.A., Müller, S., Lund, P.J., Juang, J., Adachi, K., Tse, T., Birnbaum, M.E., et al. (2015). Clonal Deletion Prunes but Does Not Eliminate Self- Specific ab CD8. Immunity 42, 929–941.

[Editors’ note: the author responses to the re-review follow.]

Reviewer #1:

*1) The observation that there are changes in the composition of the CD8 T cell pool based on memory markers remains intriguing. I am still not sure, however, whether this is a relative or an absolute change. The reasons are that the gates on the naïve populations are different for each subject (as seen in Figure 1), making the quantification of the relative number of naïve T cells seem a little random.*

We thank the reviewers for their positive assessment of our study. The gates were not perfectly identical in Figure 1 because the samples were acquired in different experiments. However, fluorescence measurements were consistent over time, standardized using CST *Check Performance*. Following the reviewer’s note, we reanalysed the same files in one single FlowJo workspace. We are now providing a sample of the resulting plots in new Figure 1, as well as new percentages (47 in HD, 35 in SVR, and 24 in cHCV instead of 43, 33 and 22 respectively). Although we recognize that acquiring all samples in a unique experiment, and applying strict gating to all samples would have been ideal, this was not possible in our prospective study. We hope the reviewer will acknowledge the minor impact of this technical limitation on the relevance of our observations.

*At the same time (if I understood the new information about absolute T cell counts correctly), absolute T cell counts were performed using different methods in HD vs SVR and cHCV patients, raising the possibility that the differences in absolute CD3 numbers as shown in Figure 1—figure supplement 2 could be a result of different methodology. These issues should be reconciled.*

Regarding the differences in absolute numbers of CD3, the reviewer is right that the methodologies used are slightly different, i.e. automatic counts on whole blood in hospital laboratory for HCV and SVR patients vs. beads-based counts on whole blood for healthy donors. However, as HCV and SVR patients were collected by leukapheresis, we did not have a way to standardize the methods. Both being well calibrated, we don’t see any reason for significant discrepancies. We have now clarified this in the manuscript. We remain confident in our conclusion of lower percentages and normal absolute counts of naïve CD8 T cells in cHCV patients, especially that the first observation is in accordance with previously published studies (Shen et al., 2010).

*2) I find it very difficult to interpret the TCR data. The cross sectional results on bulk naïve cells in Figure 1 show somewhat subtle differences that could be caused by a multitude of factors. Data on specific cells like from the one subject in Figure 2 should be more revealing. In this case, it seems clear that just one clonotype (already dominant in the naïve cells) is responsible for all memory cells. Would that not imply that one clonotype might have some cross-reactivity and thus have expanded?*

We thank the reviewer for these comments. We acknowledge that our data on bulk naïve cells are reinforced by repertoire analysis on sorted Mart1-specific naïve and memory inexperienced T cells (Figure 2). We agree with the reviewer that our observation of predominant Vβ18 clone within Mart1-specific memory-phenotype cells argues for it being the progeny of the dominant Vβ18 clone within Mart1-specific naïve-phenotype cells. Our interpretation is that the shift in the CD45RA/CD27 phenotype of inexperienced T cells is accompanied by clonal selection – and thus not a reflection of abnormal surface expression. However, we believe this does not inform us on the mechanism of expansion, i.e. specific vs. aspecific (homeostatic) differentiation, both being described as triggering clonal selection (Mikszta et al., 1999; Qi et al., 2014). We have now clarified this point in our manuscript.

*3) The analysis of specific cells for naïve versus memory population has been significantly strengthened by the new Figure 3, justifying the overall conclusion that more T cells display a memory phenotype. I have still some concerns, especially regarding the longitudinal and cross-sectional analysis of MART responses (Figure 2 and Figure 4), given that the gates on naïve cells seem to be different in every plot. How was the "right" gate determined in each case? Where the different time-points for a patient not stained in a single experiment or why do the staining patterns/signal strength look so different?*

As in point 1, the reviewer is correct that there are minor changes in naïve/memory gates (please see above for justification). With respect to Figure 2 and Figure 4, the "right" gate for tetramer-specific populations was set using naïve/memory markers on bulk CD8 T cells, acquired on the same day as given patients were analyzed.

*Regarding the cross sectional analysis in Figure 4, my concern remains that the differences seen in the few patients more than 2 years after treatment (already barely significant) might be driven by patient selection (exemplified by patients S11 and S12 with low memory percentages already pre and post treatment, potentially skewing the results for the 6 subjects included at more than 2 years post treatment). The case that treatment normalizes the phenotype remains not very strong in my opinion (did the bulk CD8 T cells show changes? more data points might be available).*

Concerning recovery after SVR, we studied all frozen material available from the five longitudinally-sampled HLA A_0201_-positive donors. Analysis on bulk CD8 T cells, as suggested by the reviewer, was conducted and is depicted in Figure 13. Although it shows an interesting pattern, we share with the reviewer the conviction that these findings will need to be confirmed in large longitudinal cohort studies. We have now added one sentence to the manuscript to indicate the need for replication.

Author response image 7.Percentages of memoryphenotype cells within bulk CD8+ T cells over time in SVR patients tested in Figure 4.**DOI:**
http://dx.doi.org/10.7554/eLife.07916.034

*4) The functional experiments using CD5 blockade remain inconclusive, given that an effect is only seen in HD. As this effect is modest and does not lead to the same level of reactivity as seen in the cHCV patients (R6) it seems unclear whether CD5 is the main mechanism here.*

We agree with the reviewer that the effect of blocking CD5 is modest and does not lead to the same level of reactivity than in cHCV patients. However, and this is consistent with reviewer 2’s interpretation, we believe the selective effect on HD is an argument for CD5 being involved in the pathogenesis underlying the hyperactivation phenotype.

*5) The in vitro expansion and ICS experiments in Figure 6 are a bit problematic. First, for each of the subjects the naïve MART precursor frequency should be given, since some of the cHCV subjects had relatively high frequencies as seen in Figure 2—figure supplement 2.*

We thank the reviewer for careful analysis of Figure 6. We now provide precursor frequencies for patients tested in IVP experiments in Figure 14. We did not observe any correlation between precursor frequency and Mart1 expansion after in vitro priming.

Author response image 8.Mart1 precursor frequencies for the HD and cHCV patients included in in vitro priming experiments in Figure 6.**DOI:**
http://dx.doi.org/10.7554/eLife.07916.035

*In any case, assuming specific cells in cHCV expand better, could this not be explained by them being memory and not naïve cells? For the functional data, the difference in granzyme B is modest and no difference is observed directly ex-vivo (Figure 6—figure supplement 3).*

The difference in granzyme B is modest but statistically significant. We have clarified this in the text.

*As for the IFN assay, are the authors really suggesting that populations requiring TAME for visualization can be detected via ICS directly ex-vivo? Detecting such small populations via ex-vivo ICS from 5 million PBMC seems unlikely.*

For ICS experiments, we stimulated 10^7^ PBMCs cells in 24-well format plates. Based on precursor frequencies (10^-5^ – 10^-6^ / CD8^+^ T cells), we expect between 2 and 20 inexperienced antigen-specific cells to be available in a given well for peptide restimulation. Given the absence of detectable IFNγ in unstimulated conditions, we believe cytokine secretion reflects a state of hyperactivation towards antigens. We therefore concluded that presumably inexperienced T cells are indeed hyperactive.

Reviewer #2:

*The authors have made numerous revisions to the manuscript, which address the major concerns I raised. The only request would be to include the data shown in Figure 11 in the manuscript. These data make, in my opinion, an important point about the impact and selectivity of the anti-CD5 blockade, and offers material support to the authors' hypothesis. This figure could be included in the supplementary material.*

We thank the reviewer for his/her interest in our study and for positive assessment of the revision. We agree that Figure 11 reinforces our hypothesis that CD5 plays a key role in lowering the threshold for TCR activation in naïve T cells from HCV patients. These results are now included in our manuscript in the Result section, and provided as Figure 5—figure supplement 4 A-D.

References:

Mikszta, J.A., McHeyzer-Williams, L.J., and McHeyzer-Williams, M.G. (1999). Antigen-driven selection of TCR In vivo: related TCR alpha-chains pair with diverse TCR beta-chains. J Immunol 163, 5978–5988.

Qi, Q., Liu, Y., Cheng, Y., Glanville, J., Zhang, D., Lee, J.Y., Olshen, R.A., Weyand, C.M., Boyd, S.D., and Goronzy, J.J. (2014). Diversity and clonal selection in the human T-cell repertoire. Proc Natl Acad Sci USA 111, 13139–13144.